# Are haloclines distributional barriers in anchialine ecosystems? Physiological response of cave shrimps to salinity

Efrain M. Chávez Solís[1], Maite Mascaro[1]*, Carlos Rosas[1], Gabriela Rodríguez-Fuentes[2], Claudia Caamal Monsreal[1], Kurt Paschke[3,4,5], Fernando Díaz[6], Denisse Re Araujo[6]

1 Unidad Multidisciplinaria de Docencia e Investigación, Facultad de Ciencias, Universidad Nacional Autónoma de México, Sisal, Yucatán, México, 2 Unidad de Química en Sisal, Facultad de Química, Universidad Nacional Autónoma de México, Sisal, Yucatán, México, 3 Instituto de Acuicultura, Universidad Austral de Chile, Puerto Mont, Chile, 4 Centro FONDAP de Investigación en Dinámica de Ecosistemas Marinos de Altas Latitudes (IDEAL), Punta Arenas, Chile, 5 Instituto Milenio Biodiversidad de Ecosistemas Antárticos y Subantárticos (BASE), Santiago, Chile, 6 Departamento de Biotecnología Marina, Centro de Investigación Científica y Educación Superior de Ensenada, Ensenada, Baja California, México

* mmm@ciencias.unam.mx

**Data Availability Statement:** All dataset files are available from the zenodo repository (https://doi.org/10.5281/zenodo.8284963).

## Abstract

Anchialine systems are coastal groundwater habitats around the world which host a unique community of cave adapted species (stygobionts). Such communities are expected to be separated by haloclines into either fresh or saline groundwater communities, hence climate changes (*e.g.*, eustatic sea level shifts) and anthropic driven changes (*e.g.*, salinization) may have a great impact on these stygobiont communities. Here we used cave-restricted species of *Typhlatya* from the Yucatan Peninsula as models to identify physiological capacities that enable the different species to thrive in marine groundwater (*T. dzilamensis*) or fresh groundwater (*T. mitchelli* and *T. pearsei*), and test if their distribution is limited by their salinity tolerance capacity. We used behavior, metabolic rates, indicators of the antioxidant system and cellular damage, and lactate content to evaluate the response of individuals to acute changes in salinity, as a recreation of crossing a halocline in the anchialine systems of the Yucatan Peninsula. Our results show that despite being sister species, some are restricted to the freshwater portion of the groundwater, while others appear to be euryhaline.

## Introduction

Groundwater ecosystems harbour essential water resources for many people and other dependent ecosystems around the globe [1], yet the rare and mostly endemic species which are restricted to these environments are poorly understood, and consistently overlooked in management strategies [2–4]. Impacts of rapid climate change and anthropogenic activities that induce modifications in the physicochemical parameters of the environment (i.e. temperature, salinity and oxygen) are expected to affect groundwater communities [5–9], however, the impacts of these changes on groundwater species vary among species and the particular

**Funding:** This study was funded by the Dirección General de Asuntos del Personal Académico, Universidad Nacional Autónoma de México, through grant PAPIIT IN203022 awarded to Dr. Carlos Rosas and grant PAPIIT IN228319 awarded to Dr. Nuno Simoes [https://www.unam.mx/]. Efrain M Chávez Solís received a postdoctoral fellowship (No. 545211) from CONACYT [https://conahcyt.mx/]. The funders had no role in study design, data collection and analysis, decision to publish, or preparation of the manuscript.

**Competing interests:** The authors have declared that no competing interests exist.

environmental change itself, which makes accurate projections for the future very difficult [4, 6, 10–13].

Anchialine ecosystems (Box 1) in the Yucatan Peninsula (YP) are considered seasonally stable, with environmental changes occurring at broader timescales (such as glacial and interglacial epochs). However, haloclines in anchialine environments are dynamic in depth, and changes in salinity within one system may vary markedly when located closer to the coast [14]. Particularly, tropical storms have rapid and profound effects on the geochemistry of anchialine systems, as they impact oxygen availability, modify the salinity, and have long term (months) impacts the carbon cycle [15]. Such changes have a significant impact on habitat availability and physiological stress, which determine the distribution of species within these environments [12, 16, 17]. Because of the limited dispersal capacities of cave-restricted aquatic species (stygobionts), their evolutionary history is tied to the dynamics of these coastal aquifers [18, 19]. Thus, the species populations we presently observe are those that, through natural selection, have developed the plasticity to withstand the environmental conditions characterising these caves. Understanding the physiological capabilities of stygobiont species today allows to recognize their capacity to endure past and future environmental fluctuations. This understanding is paramount in the face of the prevailing anthropic trends of groundwater use and rapidly advancing climate changes (e.g., salinization due to over-extraction of fresh groundwater, temperature increase and sea level rise). Acute changes in salinity are one of the defining characteristics of anchialine environments [20], and salinity is a major environmental factor that determines the distribution and composition of aquatic communities [21]. Consequently, the stygobiont community within anchialine ecosystems is separated vertically by haloclines, into distinct fresh and marine groundwater sectors within the subterranean habitat [22–25].

## Box 1. Glossary

| | |
|---|---|
| Anchialine ecosystems | Globally dispersed coastal groundwater habitats, occurring in karstic and volcanic geological settings, which are characterized by drastic changes in salinity either temporarily (daily) or spatially (vertically stratified), hosting fresh, brackish and marine groundwater due to the mixing of terrestrial-borne freshwater and the intruding seawater [20, 31, 32]. |
| Metabolic Rate | When measured as oxygen removal from water it is actually oxygen uptake [33]. Oxygen uptake reflects the speed at which aerobic metabolism produces ATP, a measure of the amount of energy required (and supplied) by the organism, which is influenced by several internal (activity level, developmental stage, body size, food intake, etc.) and external factors (temperature, salinity, hypoxia, etc.). See Chabot et al. [34] |
| Aerobic Scope | The energy that an individual can produce through aerobic respiration that is beyond the metabolic cost of basal processes and is intended to perform ecological functions [35, 36]. |
| *Pejus* | Latin for "getting worse". Represents a moderate stress range where the maintenance costs increase and AS is reduced [35, 37]. |
| *Pessimum* | Extreme stress condition where the requirement for essential maintenance processes uptakes all available energy, exceeding the aerobic capacity to deliver such ATP demand, which is consequently partially fueled by anaerobic metabolism, reducing the AS to zero. |
| *Lethal* | The energy requirement for essential processes is greater than what the organism can produce, it results in a negative AS and ultimately the death of the organism. |

Species of the *Typhlatya* genus (Atyidae) have a globally disjunct distribution in coastal aquifers [26, 27], and are also distinctly distributed throughout the YP in either the fresh groundwater (FG) or the marine groundwater (MG) [17, 28–30]. However, some species have been reported in both FG and MG [29], and recent observations have documented individuals

traversing a halocline (S1 Video). These observations lead to ecological, physiological, and evolutionary inquiries: Do all *Typhlatya* species within the YP anchialine environments possess the capacity to navigate through a halocline? What is the metabolic cost of such behaviour? Could the recently reported variations in aerobic scope (glossary Box 1) contribute to this capacity? Moreover, how might such observations affect their vulnerability considering climate change projections?

This study aimed to assess the physiological response of three *Typhlatya* species of the anchialine systems of the YP to acute changes in salinity. We designed experiments to measure and compare behavioural, physiological (routine metabolic rate) and biochemical (antioxidant enzymes, oxidative damage and anaerobic metabolism) indicators in *T. mitchelli*, *T. pearsei* and *T. dzilamensis* subject to water conditions that resembled changes in salinity consistent with crossing the halocline.

## Materials and methods

### Animal collection

*Typhlatya mitchelli*, *T. pearsei* and *T. dzilamensis* were manually collected using SCUBA cave diving techniques from cenotes Tza Itza (20.730311° N, 89.46608° W), Nohmozon (20.623266°N, 89.384207° W) and the Ponderosa System (entering through cenote Xtabay 20.499183° N, 87.260842° W), respectively. The Ponderosa System is an anchialine system with the Xtabay cenote at approximately 2 km inland from the north-eastern coast of the YP, with a vertical stratification of salinity and temperature. Cenotes Tza Itza and Nohmozon are on the northwest YP approximately 80 km inland, and are exclusively freshwater sites with a thermally homogeneous water column (see Fig 1 in [16]).

The collection of *T. pearsei* was conducted in night dives (as they are absent during the day) in the exclusively freshwater cenote pool of Nohmozon, from depths between 4 and 16 m. *T. mitchelli* individuals were collected from the exclusively fresh groundwater cenote Tza Itza and from the fresh groundwater portion of the "Repair Shop" cavern (Ponderosa system). For the latter, divers navigated 400 m (20–25 minutes) underground from the Xtabay Cenote through the Ponderosa System to reach the cavern area of the Repair Shop cenote. Overall, divers could spend 20–30 minutes searching and capturing *T. mitchelli*, which were found at depths between 2 and 8 m. *T. dzilamensis* were collected from the underlaying MG in the cave adjacent to Xtabay cenote (Ponderosa system), at depths ranging from 13.5 to 15.5 m. Divers collected animals using a hand net and individually placed them in 50 ml falcon tubes. All collections and species identifications were led by EC with the support of certified cave-divers.

### Salinity scenarios

To examine the tolerance to salinity in *T. mitchelli*, *T. pearsei* and *T. dzilamensis*, we defined three contrasting scenarios based on the water conditions in which these species have been observed [16, 26, 29]. Salinity at collection site was defined as "native salinity": 0.7 $S_p$ for *T. pearsei* from cenote Nohmozon, 3 $S_p$ ($S_a$ 3.01417 g/kg) for *T. mitchelli* from the uppermost groundwater layer in Ponderosa System, and 35 $S_p$ ($S_a$ 35.16531 g/kg) for *T. dzilamensis*, from the bottom groundwater layer in Ponderosa System. An extreme salinity scenario was established at 3 $S_p$ for *T. dzilamensis* and 35 $S_p$ for *T. pearsei* and *T. mitchelli*, as these were the lowest and highest salinities measured in the Ponderosa system, respectively. A scenario of "moderate" salinity was established at 14 $S_p$ for all species, considering this as the intermediate salinity between both water masses.

Marine groundwater was taken from below the halocline in Ponderosa (35 $S_p$) using 0.5-gallon plastic bags. Intermediate salinity water was prepared at 14 $S_p$ mixing marine

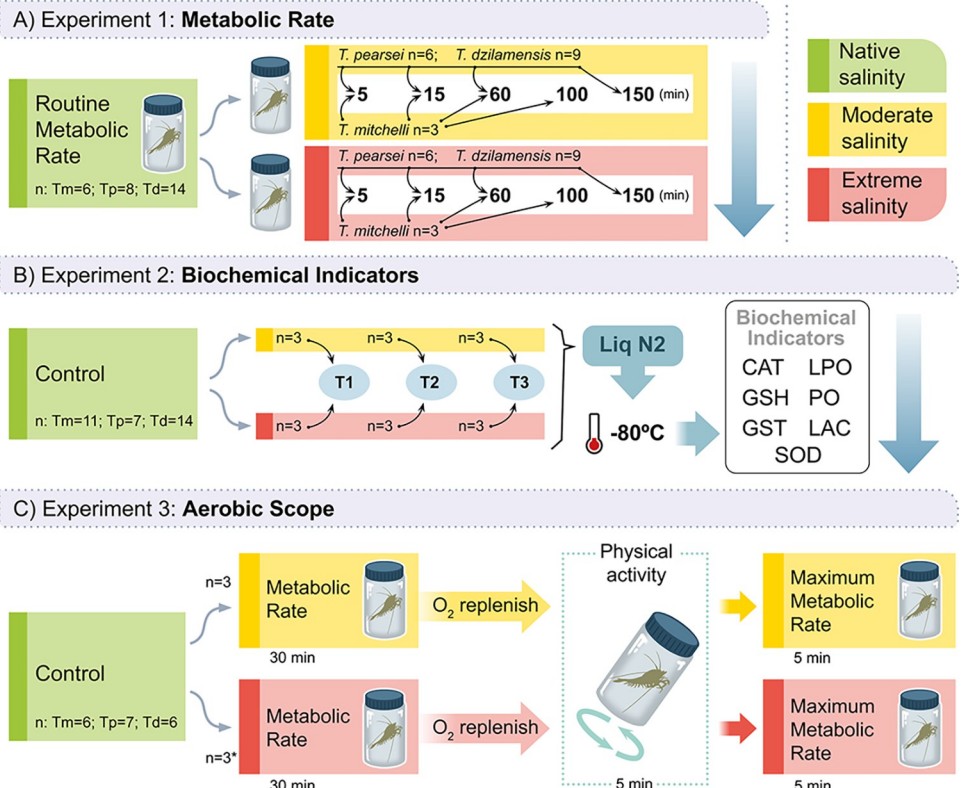

**Fig 1. Graphic summary of experiments.** (A) routine metabolic rate (RMR), (B) biochemical indicators of anaerobic metabolism, antioxidant system and cellular damage and (C) aerobic scope of three *Typhlatya* species were measured in their native salinity (green boxes: 0.7, 3 and 35 $S_p$ for *T. mitchelli*, *T. pearsei* and *T. dzilamensis*, respectively) and after being subject to either a moderate (yellow boxes: 14 $S_p$ for all species) or an extreme salinity challenge (orange boxes: 35 $S_p$ for *T. mitchelli* and *T. pearsei*, and 3 $S_p$ for *T. dzilamensis*). The number (n) of individuals of *T. mitchelli* (Tm), *T. pearsei* (Tp) and *T. dzilamensis* (Td) used as independent replicates in each case is given. Exposure to salinity in (A) and (B) had a total duration of 100 (for *T. mitchelli*) or 150 minutes (*T. pearsei* and *T. dzilamensis*). RMR in (A) was measured after 5, 15, 60, 100 and 150 minutes of exposure and biochemical indicators in (B) were measured in samples of *T. mitchelli* taken after 7, 15 and 80 minutes, and of *T. dzilamensis* and *T. pearsei* taken after 15, 60 and 150 minutes of exposure (T1-T3). Individuals were frozen in liquid nitrogen on site and stored at -80˚C until analysis. RMR in (C) was measured during 30 minutes of salinity exposure and subtracted from maximum metabolic rate (MMR) measured during an additional 5 minutes in the same individuals after 5 minutes of intense physical activity had been induced. RMR and MMR were used to calculate aerobic scope of all three species (except *T. mitchelli* at the extreme salinity*).

groundwater with surface water from either Cenote Xtabay for trials with *T. mitchelli* and *T. dzilamensis*, or from Cenote Nohmozon for trials with *T. pearsei*. For trials with *T. pearsei*, marine groundwater was transported to Nohmozon from Ponderosa in 20 l containers with an aerator one day prior to experiments.

Vertical water column profiles were obtained using a Hydrolab DS5 and published in Chávez-Solís et al. [16]. Additionally, salinity from all the collected groundwater was confirmed with a refractometer prior to all trials.

## Methodological background

The hypotheses of energy-limited tolerance to stress [38] and the oxygen- and capacity-limited thermal tolerance [39] constitute an integrating approach to address the response of individuals to stress through different organizational levels, *i.e.*, from molecules to whole organisms, populations, and species. The impact of stress on metabolism and energy allocation has been

thoroughly studied through thermal biology and bioenergetics, and the ranges of stress in individuals as they recede from the *optimum* are termed *pejus*, *pessimum*, and *lethal* (Box 1; [40, 41]). Measuring the speed at which aerobic metabolism produces ATP (metabolic rates) and its fluctuations under different environmental conditions shows the effect of such fluctuations on aerobic scope (the energy that an individual can produce beyond the metabolic cost of basal processes which is used to perform ecological functions; AS [35, 36]) as a reflection of the metabolic limit to an individual's tolerance [35, 38, 40–42].

Because AS relates to all aerobic activities (e. g. locomotion, digestion, reproduction, and competition), it has been used as a proxy of physiological performance [36, 43]. The AS is maximal at optimal environmental conditions [35, 40, 44]. At suboptimal conditions, however, energy allocation is prioritized to maintenance processes, fitness-related activities are impaired, and the aerobic scope decreases. Negative AS values are only possible for short periods of time, as this implies that life maintenance costs exceed the ATP production [35, 40, 45]. Maintaining a positive AS is necessary for the long-term survival of the organisms and their populations [38].

When the aerobic metabolism increases due to an increase in energy demand, there is an elevated production of reactive oxygen species (ROS) [46]. To avoid damage from oxidative stress, organisms rely on the antioxidant system (AOS) [46, 47], a series of oxidation-reduction reactions that maintain cells in a redox equilibrium by donating electrons to the free radicals and managing the less reactive or detoxified molecules there off [48, 49]. Changes in antioxidant biomarkers (such as catalase, superoxide dismutase, glutathione, etc.) reflect activation of the antioxidant system, whereas indicators of cellular damage (such as protein carbonylation and lipid peroxidation) reflect an AOS overwhelmed by ROS, [48, 50–52].

We used measures of the routine metabolic rate (RMR), aerobic scope (AS), and biomarkers of the antioxidant system, anaerobic metabolism and cellular damage to characterise the response of the three *Typhlatya* species to an abrupt change from their native condition to either a moderate or an extreme salinity.

## Experiment 1: Metabolic rates at different salinity scenarios

We used individual oxygen uptake during normal activity in a closed respirometer as a measure of the RMR [33, 53]. Experiments to measure RMR were conducted *in situ* shortly after collection (30 to 45 minutes). Individuals collected in FG were kept in perforated 50ml falcon tubes underwater in the cenote pool, to allow the flow of water while avoiding changes in temperature and dissolved oxygen. Those collected in MG were kept in closed 50 ml falcon tubes in the cenote pool (which is FG) to avoid changes in salinity and temperature. Once the experimental salinity scenarios were ready, each individual was extracted using a net and placed into 17 ml clear glass respirometer chambers containing water either in the native or experimental salinity. Oxygen uptake rates were then measured using 5 mm spot oxygen sensors glued to the inside of the chambers and coupled to a Witrox4 amplifier through an external optical fiber cable (Loligo systems, Denmark). Two respirometer chambers were kept without individuals in each trial to account for background respiration (control). The rack containing experimental and control chambers was kept underwater to avoid temperature changes during the experiments. Electrical instruments were connected to a portable solar panel battery (GoalZero Yeti 1250) or a gasoline electric generator (Honda Inverter EU 2000i), located outside the cenote gallery at each study site. Oxygen uptake was calculated using the following equation:

$$VO_2 = \frac{\delta PO_2 * V_r - C}{T * W}$$

where $VO_2$ is the oxygen consumption of each individual (expressed in $mgO_2\ hr^{-1}\ g^{-1}$); $\delta PO_2$ is the slope of oxygen concentration decrease over time, obtained from 5-minute intervals (expressed in $mgO_2\ sec^{-1}$); $V_r$ is the volume in the chamber, calculated as the total volume of the chamber (L) minus the mass of each individual (g) (i.e. animal density was assumed to be equal to that of water); C is the mean slope of oxygen reduction in control chambers (*i.e.* background respiration; $mgO_2\ seg^{-1}$); T is the 5-minute intervals (hour); and W is mass of the individual (g).

When *Typhlatya* were under their native salinity, we took the maximum slope of oxygen consumption in 150 minutes; when transferred to challenging salinity conditions, we took the maximum slope in several 5-minute intervals throughout exposure [36, 45]. The selection of these intervals was done *post hoc* on the basis of a visual inspection of the complete data series and considered distinct behavioural components displayed by the *Typhlatya* (such as, loss of balance, spasms and tail-flips) linked to clear changes in oxygen uptake. The RMR of *T. mitchelli* during the salinity challenge was measured at 5, 15, 60 and 100 minutes, whereas that of *T. pearsei* and *T. dzilamensis* was measured at 5, 15, 60 and 150 minutes after being transferred to target salinities (Fig 1A). The inspection in real time of these behavioural cues also allowed to define the maximum duration of exposure at 100 minutes for *T. mitchelli* for both moderate and extreme salinity trials, and 150 minutes in those for *T. dzilamensis* and *T. pearsei*.

Of the 12 *T. mitchelli* collected from Tza Itza and Ponderosa System, 6, 3 and 3 individuals were used to measure the RMR in the native, intermediate and extreme salinities, respectively. Of the 20 *T. pearsei* collected from Nohmozon, 8, 6 and 6 were used to measure the RMR in the native, intermediate and extreme salinities, respectively. Of the 32 *T. dzilamensis* from Ponderosa System, 14, 9 and 9 were used to measure the RMR in the native, intermediate and extreme salinities, respectively. Every individual of *Typhlatya* was used only once throughout RMR trials.

Once RMR trials were terminated, all individuals were dried, weighed, and fixed in liquid nitrogen. On arrival at the laboratory, samples were transferred to a deep freezer and kept at -80°C until biochemical analyses were conducted.

## Experiment 2: Biochemical indicators

The activity of catalase (CAT), glutathione S-transferase (GST), total glutathione (GSH) and superoxide dismutase (SOD) were used as indicators of the activation of the antioxidant system, whereas protein carbonylation (PO) and lipid peroxidation (LPO) were used as indicators of oxidative damage [51, 52, 54, 55]. Lactate concentration (LAC) was used as an indicator of anaerobiosis [56].

Behavioural observations together with results of the RMR trials served as a guideline to establish three critical moments that could reveal changes in the antioxidant system, anaerobic metabolism and cellular damage (if any) in response to salinity. Three exposure times were established at ($T_1$) 7, ($T_2$) 15 and ($T_3$) 80 minutes for *T. mitchelli*, and ($T_1$) 15, ($T_2$) 60 and ($T_3$) 150 minutes for *T. pearsei* and *T. dzilamensis* (Fig 1B). Nine individuals of each species were abruptly transferred to the respective target salinities and groups of three individuals were sampled and frozen in liquid nitrogen after each exposure time. Individuals of *T. pearsei* and *T. dzilamensis* that were deep-frozen on the termination of Experiment 1 were added to increase sampling effort at $T_3$. In addition, 11, 7 and 14 individuals of *T. mitchelli*, *T. pearsei* and *T. dzilamensis*, respectively, were collected from sites with their native salinity and immediately frozen to serve as a baseline (controls). All samples were transferred to the laboratory and stored at -80°C.

Each individual sample was processed in an assay tube immersed in ice using a Potter-Elvehjem homogenizer with a PTFE pistil. PO, LPO and GSH were analyzed following procedures outlined in Rodríguez-Fuentes et al. [57], where Tris buffer at a concentration of 0.05M (pre-set crystals pH7.4 of Tris [hydroxymethyl] aminomethane and Tris HCL) was added to a proportion of 1 ml buffer for every 50 mg of wet tissue. SOD, CAT and GST were analyzed from the supernatant after centrifuging the rest of the homogenate at 10,000 rpm during 5 minutes at 4°C. GSH and SOD were determined using a Sigma kits CA0260 and 19160, respectively, following the manufacturer's indications. GST enzyme activity was using a Sigma CS04 kit with CDNB as substrate for spectrophotometric measurement at 412 nm every 15 sec during 5 min, as described in Ellman et al. [58] and Habig et al. [59]. CAT activity was determined by measuring the reduction rate of $H_2O_2$ at 405 nm upon reaction with ammonium molybdate, as described in Góth [60], modified by Hadwan and Abed [61]. Total protein was measured following the adaptation to microplate of the Bradford method [62], which was developed by personnel from the Ecotoxicology laboratory of the Faculty of Chemistry, UNAM, in Sisal, Yucatan, Mexico. PO was determined following Mesquita [63]. The FOX method was used to quantify LPO using the Sigma peroxidetect kit (PD1). LAC was determined with TRINITY and Pointe Scientific kits following the manufacturer's indications.

All biochemical determinations were performed with duplicate subsamples in the Ecotoxicology laboratory of the Faculty of Chemistry, UNAM, in Sisal, Yucatan, Mexico. Negative spectrophotometric absorbances of both duplicates were considered as not detected.

## Experiment 3: Aerobic scope

The absolute aerobic scope (AS) of the three *Typhlatya* species was estimated as the difference between maximum metabolic rate (MMR) and RMR, used here as a proxy of the capacity for oxygen transport above the energy expenditure required by the salinity change [64]. The AS of individuals of the three *Typhlatya* species in native (control) and challenging salinities were obtained.

To calculate the RMR during the salinity trials, three individuals of each species were placed in individual respirometry chambers (as described earlier) with water salinity corresponding to each challenge and their oxygen uptake was measured for 30 min. Water in the chambers was then replaced with oxygen-saturated water of the same salinity and the same individuals were submitted to 5 minutes of physical activity. This was achieved using an automatic device that rotated the chambers at 25 revolutions per minute in a twirling movement that induced shrimps to swim constantly in the chamber's relatively reduced space. Immediately after physical activity, the chambers were transferred to the underwater rack for an additional 5-minute period of respirometry measurements, which corresponded to the MMR of each individual (Fig 1C). All individuals were finally weighed, and AS results were expressed in $mgO_2 \, hr^{-1} \, g^{-1}$. Individuals of *Typhlatya* used in this experiment were different than those in Experiment 1.

## Statistical analysis

Permutational tests of hypotheses were used to analyse both the univariate RMR and lactate data and the multivariate antioxidant defence and oxidative damage data. We preferred this approach because response variables, in general, failed to conform to the normal distribution; and the antioxidant defence data had strong covariation patterns, a feature that further restricts compliance to multivariate normality [65]. Permutational procedures are based on generating an empirical distribution of the statistic (*F* or *t* in this case) under the null hypothesis, and assessing the cumulative frequency of values as large as or greater than the observed value (often called pseudo-*F* or pseudo-*t*; [66]. Empirical distributions were obtained using

9999 permutations of raw data in one-way models and the same number of permutations of the residuals under the reduced model in two-way full factorial models [66].

**Analysis of the routine metabolic rate (RMR).** To compare the RMR (response variable) between species under their native salinity, we first conducted a one-way permutational ANOVA with species as a fixed factor (explanatory variable with 3 levels: *T. mitchelli*, *T. pearsei*, *T. dzilamensis*). To examine changes in RMR in individuals through time of exposure to a moderate or an extreme change in salinity, we used a full two-factor ANOVA with time (with 4 levels: $T_1$, $T_2$, $T_3$ and $T_4$) and magnitude of the salinity challenge (with 2 levels: moderate and extreme) as fixed factors (explanatory variables). Because the time span of the response to salinity among the three *Typhlatya* was not comparable, ANOVAs were applied to each species separately. Significant interaction terms (permutational $p < 0.05$) would reflect that changes in RMR through time followed a different pattern depending on the magnitude of the salinity challenge. When the interaction term or the main term related to time resulted significant, group means were further compared using permutational univariate *t* tests [66]. RMR data was square root transformed prior to the calculation of the resemblance matrixes of Euclidean distances between samples. Boxplots on untransformed RMR data were used to visualize variation among species, time and salinity challenges.

**Analysis of the antioxidant system and oxidative damage (AOS).** Principal Coordinate Analysis (PCoA) were used to evaluate changes in the antioxidant system and oxidative cellular damage of *Typhlatya* species. Multivariate datasets with the results of the six response variables CAT, GST, GSH, SOD, PO and LPO (henceforth collectively referred to as biochemical indicators BI) for each species were square root transformed and normalised prior to the calculation of the resemblance matrixes of Euclidean distances between samples. Eigen analysis applied on the resemble matrixes calculated the percentage of the total variation contained in the first 2-dimensions of each configuration (PCO1 and PCO2), whereas the PCoA scores allowed to order samples (individuals) in a space of reduced dimensionality while preserving their distance relationships as well as possible [67].

Firstly, a PCoA was applied to BI measured in individuals of the three species of *Typhlatya* under native salinity conditions. A one-way permutational MANOVA served to evaluate statistical differences in BI (response variables) between species as a fixed factor (explanatory variable). Secondly, three PCoAs were applied to BI measured before and after exposure to a challenging salinity in each species separately. Each of these were accompanied by a two-way permutational MANOVA to examine whether BI varied significantly before (native salinity condition) and throughout the time of exposure (with 3 levels: $T_1$, $T_2$ and $T_3$) depending on the magnitude of the salinity challenge (with 2 levels: moderate and extreme). Because baseline indicators of species maintained at their native salinity had no temporal dimension, these were included in the model as a hanging control and compared to all non-native measures accordingly [68]. Here again, a statistically significant interaction term would indicate that changes in BI through time followed a different pattern depending on the magnitude of the salinity challenge. When the interaction term or the main term related to factor time resulted significant, group means were further compared using permutational multivariate *t* tests [66].

**Analysis of lactate content.** Changes in lactate both between species and before and throughout the salinity challenge were analysed using the same underlying models described for the antioxidant system. As with RMR, however, the resemble matrixes of Euclidean distances were calculated between samples considering only one response and boxplots of untransformed data were used to visualize its variation.

Exploratory analysis and boxplot charting were conducted using the ggplot2 library [69] in R [70]. Principal Coordinate Analysis and permutational tests of hypothesis were performed with PRIMER 7 [71] and the PERMANOVA add-in [68].

**Analysis of aerobic scope.** The reduced number of individuals that were available for the aerobic scope experiment, together with the elimination of data points due to inconsistent $O_2$ readings (before or after exercise), resulted in an unbalanced and insufficient sample size to conduct hypotheses testing, hence, results of Experiment 3 were analysed using descriptive statistical tools.

## Results

### Metabolic rates and behavior during salinity challenges

Mean routine metabolic rates (RMR) of *Typhlatya* species under their respective native salinities did not differ significantly (Table 1). Furthermore, the RMR of all three species increased immediately after individuals were transferred to both a moderate and an extreme condition. However, both the magnitude of the increase and the time to reduce the RMR thereafter was different among species (Fig 2).

Changes in the RMR of *T. mitchelli* through time varied depending on the magnitude of the salinity challenge (Table 1). After 5 minutes of exposure to 14 $S_p$, the RMR had increased to a significantly higher value compared to all other moments of exposure. After 15, 60 and 100 minutes, *T. mitchelli* decreased its RMR gradually, reaching almost zero at the end of the trials

**Table 1. Results of permutational ANOVAs on the routine metabolic rate (RMR) measured in *T. mitchelli*, *T. pearsei* and *T. dzilamensis* at four moments ($T_1$, $T_2$, $T_3$ and $T_4$) after being exposed to either a moderate or an extreme salinity challenge.** Control RMR were measured in individuals of each species at their native salinity and compared among species. See descriptions in the text for references to the time of exposure and water salinity used to challenge each species. The *pseudo-F* and *p* values for each source of variation is given together with the results of post hoc comparisons of mean RMR values every time the main terms "Species", "Time" or the interaction term were significant; different letters represent mean values that were statistically distinguishable ($p < 0.05$).

| General MANOVA | | | Groups of means following post-hoc tests | | | |
|---|---|---|---|---|---|---|
| **All species native salinity (controls)** | | | | | | |
| **Source** | ***pseudo-F*** | ***p*** | | | | |
| **Species** | **0.09** | **0.095** | | | | |
| ***T. mitchelli*** | | | | | | |
| **Source** | ***pseudo-F*** | ***p*** | | | | |
| Salinity challenge | 1.24 | 0.2738 | moderate-5 min | a | | |
| Time | 26.27 | < 0.001 | moderate-15 min | | b | |
| Interaction | 4.96 | < 0.01 | moderate-60 min | | b | c |
| | | | moderate-100 min | | | c |
| | | | extreme-5 min | | | |
| | | | extreme -15 min | | | |
| | | | extreme -60 min | | | |
| | | | extreme -100 min | | | |
| ***T. pearsei*** | | | | | | |
| Source | *pseudo-F* | *p* | | | | |
| Salinity challenge | 11.90 | < 0.001 | time 1 | a | | |
| Time | 21.38 | < 0.001 | time 2 | a | | |
| Interaction | 2.80 | 0.0504 | time 3 | | b | |
| | | | time 4 | | | c |
| ***T. dzilamensis*** | | | | | | |
| Source | *pseudo-F* | *p* | | | | |
| Salinity challenge | 1.25 | 0.2751 | time 1 | a | | |
| Time | 15.30 | < 0.001 | time 2 | | b | |
| Interaction | 1.04 | 0.3851 | time 3 | | | c |
| | | | time 4 | | | c |

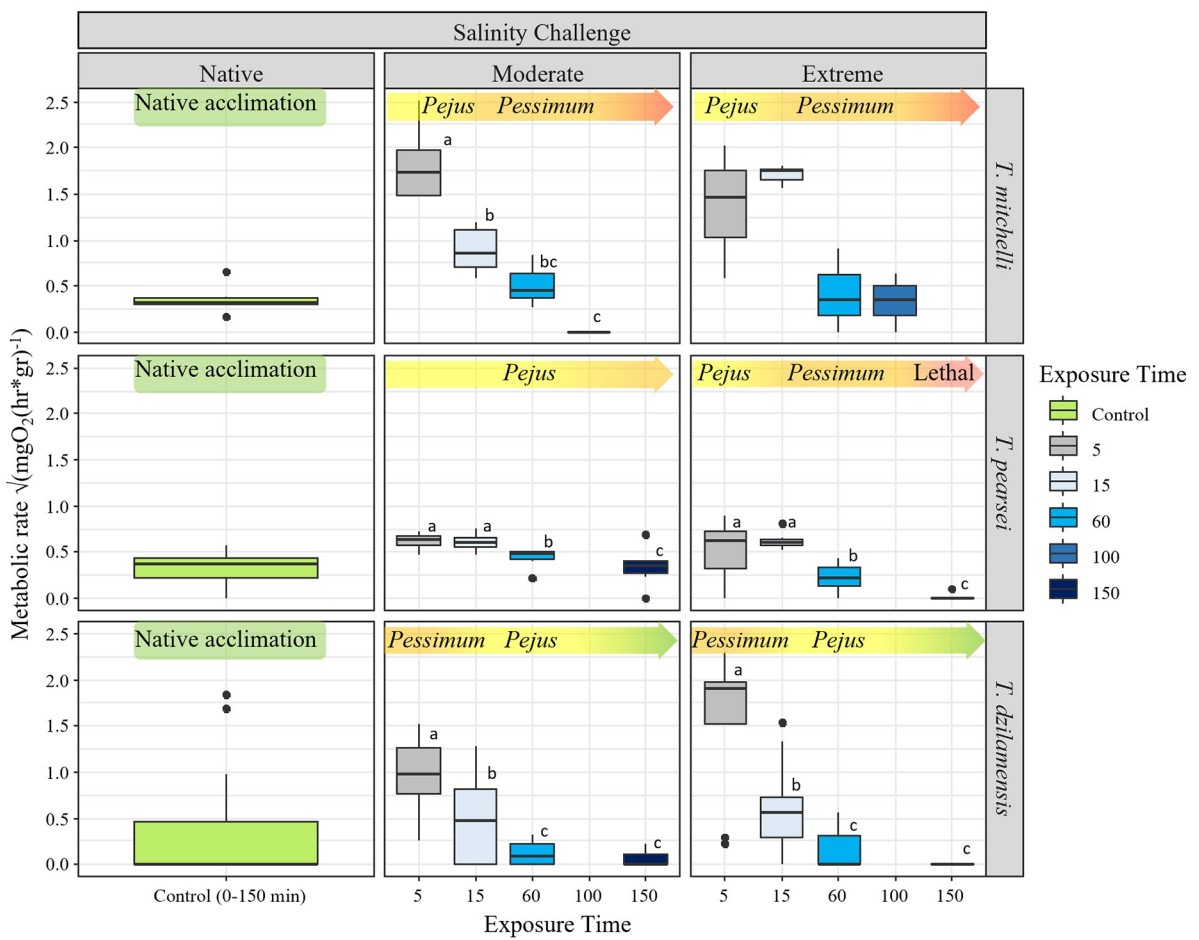

**Fig 2. Routine metabolic rates (RMR) of *T. mitchelli*, *T. pearsei* and *T. dzilamensis* exposed to moderate (14 $S_p$) and extreme salinity changes (35 $S_p$ for *T. mitchelli* and *T. pearsei*, and 3 $S_p$ for *T. dzilamensis*) during 100 or 150 minutes.** Green boxplots indicate the RMR measured when each species was subject to their native salinity (0.7 $S_p$ for *T. mitchelli* and *T. pearsei*, and 35 $S_p$ for *T. dzilamensis*). Values are square root-transformations of oxygen uptake ($\sqrt{[mgO2(lt*hr*gr)-1]}$). Stress range indicators at the top of each facet are based on the dynamic energy budget and the Energy-limited tolerance to stress concepts, which are represented with gradient bars as: native acclimation (green), *pejus* (yellow), *pessimum* (orange) and *lethal* (pink). Different letters above each boxplot indicate significant differences.

(Table 1 and Fig 2; dataset available at https://doi.org/10.5281/zenodo.8284963). In contrast, the RMR of *T. mitchelli* under 34 $S_p$, increased to values as high as in the moderate challenge until the first 15 minutes had elapsed; it decreased after 60 minutes of exposure; and attained values even lower than those under their native salinity after 100 minutes had elapsed (Fig 2). Post hoc comparisons in this case, however, were not able to detect statistical differences between moments in time (Table 1), probably due to small sample size and the relatively greater dispersion amongst these measures.

These results may explain the restlessness and tail-flips observed in some *T. mitchelli* as soon as salinity transfer occurred, as well as the spasms and loss of equilibrium presented by other individuals. The gradual but clear reduction in RMR also corresponds with individuals remaining motionless during the rest of the trials. It is important to note that all *T. mitchelli* survived despite the markedly low RMR values registered at the end of the trials (Fig 2).

When abruptly exposed to both moderate and extreme salinity changes, *T. pearsei* increased its RMR to rates lower than both *T. mitchelli* and *T. dzilamensis* (Fig 2). Mean RMR was overall

significantly higher in moderate than extreme salinity (Table 1 and Fig 2). It also significantly decreased as exposure time elapsed but did so in a similar way irrespective of the magnitude of the salinity challenge (as indicated by the non-significant interaction term; Table 1). After 150 minutes of exposure to either 14 or 34 $S_p$, the RMR had dropped to values similar or below those measured at their native salinity (Fig 2). Several *T. pearsei* individuals presented tail-flips when transferred to both the moderate and the extreme salinity challenges. But, individuals would swim around the respirometry chamber without displaying spasms or loss of balance. Only one of the six individuals of *T. pearsei* survived after 150 minutes of exposure to 34 $S_p$.

*T. dzilamensis* increased its RMR to similar overall values irrespective of the magnitude of the change in salinity (Table 1 and Fig 2). The gradual decrease in metabolic rate through time was also similar after being transferred to either 14 or 34 $S_p$. Mean RMR values were significantly lower starting at 5 and until reaching 60 minutes of exposure, but were statistically similar thereafter (Table 1). Behaviour of *T. dzilamensis* contrasted markedly with that of its conspecifics: individuals did not show stress in the form of tail-flips, spasms or loss of balance, and their swimming activity was similar to that observed under native salinity conditions. Although RMR values were close to zero after 150 minutes in both salinities (Fig 2), all *T. dzilamensis* survived until the end of trials.

## Antioxidant system

The principal coordinate analysis (PCoA) applied on the antioxidant system and oxidative damage in this study resulted in four ordinations all of which had approximately 55% of the total variation explained in the first (horizontal; PCo1) and second (vertical; PCo2) principal coordinates (Fig 3A–3D). A third coordinate (depth; PCo3) contributed with another 18.6% of the total variation (S1 Table and S1 Fig). Untransformed values of the antioxidant system and oxidative damage are included in the dataset (https://doi.org/10.5281/zenodo.8284963).

## Comparison between species

Ordination of the antioxidant enzymes and oxidative damage in the *Typhlatya* species under native conditions showed that SOD and GST were those that most contributed to the separation of individuals in the horizontal axis (S1 Table). *T. mitchelli* individuals showed the highest activity of these enzymes (on the left-hand side of the ordination map; Fig 3A), followed by *T. pearsei* and *T. dzilamensis* with the lowest activity (on the right-hand side). CAT, GSH and PO contributed to separate samples on the vertical axis, where *T. mitchelli* (with CAT) and *T. dzilamensis* (with GSH) had higher values than *T. pearsei*. LPO only contributed to the separation of samples in the third principal component, but this oxidative indicator had similar mean values in all species (S1 Table).

In summary, *T. mitchelli* showed higher activity of SOD, GST and CAT under native conditions, while *T. dzilamensis* had the lowest activity in the first two enzymes but showed high GSH activity. *T. pearsei* showed intermediate activity of antioxidant enzymes and oxidative damage indicators (Fig 3A). Results of the permutational MANOVA showed that the differences described above were statistically significant when comparing *T. dzilamensis* from *T. mitchelli*, but *T. pearsei* was not significantly distinguishable from its other two conspecifics (Table 2).

## T. mitchelli

The ordination of the antioxidant system and oxidative damage indicators in *T. mitchelli* showed that SOD and GST were the two indicators that contributed most to separate the control (with high enzyme activity on the left) from the challenged individuals (with low activity

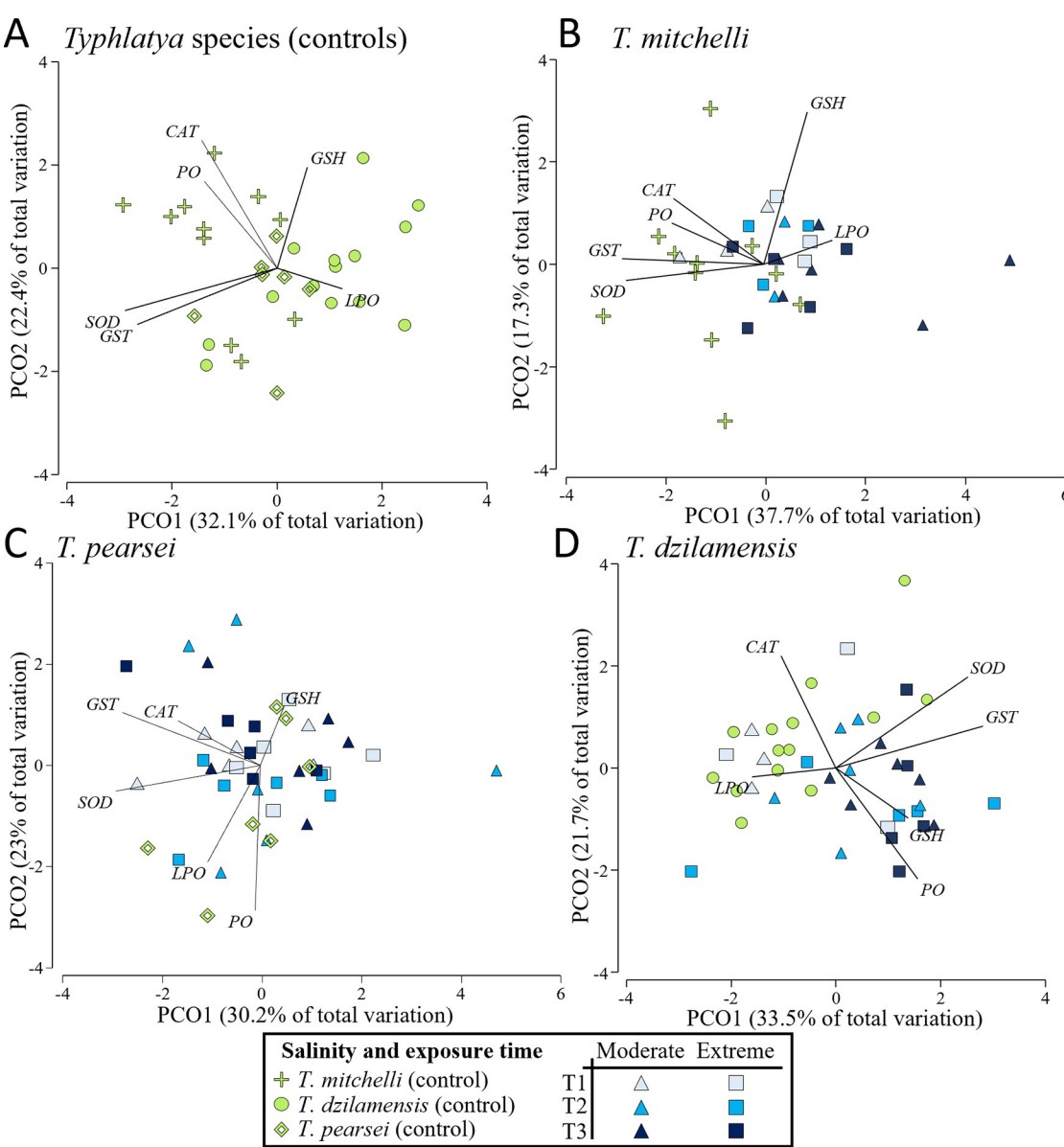

**Fig 3. Principal coordinate analysis of the antioxidant and cell damage indicators.** Principal Coordinate Analysis (PCoA) on six indicators of the antioxidative system and oxidative damage measured under control conditions in individuals of each species collected directly from their native salinity (A), and *T. mitchelli* (B), *T. pearsei* (C) and *T. dzilamensis* (D) comparing native salinity conditions with those after three moments (T₁, T₂ and T₃) of exposure to either a moderate or an extreme salinity challenge. Exposure times were 7, 15 and 80 minutes for *T. mitchelli*; and 15, 60 and 150 minutes for *T. pearsei* and *T. dzilamensis*.

on the right-hand side of the horizontal axis; S2 Table and Fig 3B). A notable difference between treated and control individuals was that the former were strikingly homogeneous irrespective of time and salinity, as shown by the low dispersion of all samples gathered at the centre of the configuration (Fig 3B). Results of the permutational MANOVA paralleled these results, since under its native salinity, *T. mitchelli* differed significantly from all experimental treatments (Table 2), suggesting that an abrupt change in salinity had a strong impact on the oxidative balance in this species. The antioxidant response to changes in salinity was similar both for the moderate and the extreme challenge and values were kept statistically unvarying

**Table 2. Results of permutational MANOVAs on six indicators of the antioxidative system and oxidative damage measured in *T. mitchelli*, *T. pearsei* and *T. dzilamensis* under native salinity conditions and at three moments (T₁, T₂ and T₃) after being exposed to either a moderate or an extreme salinity challenge.** Hanging controls were measured in individuals of each species collected directly from their native salinity. See descriptions in the text for references to the time of exposure and water salinity used to challenge each species. Antioxidative enzymes were: catalase (CAT), glutathione S-transferase (GST), total glutathione (GSH), and superoxide dismutase (SOD). Oxidative damage was quantified through: protein carbonylation (PO) and lipid peroxidation (LPO). The pseudo-*F* and p values for each source of variation is given together with number of unique permutations obtained for the corresponding test; different letters represent mean values that were statistically distinguishable ($p < 0.05$).

| General MANOVA | | | Groups of means following post-hoc tests | | | |
|---|---|---|---|---|---|---|
| **All species native salinity (controls)** | | | | | | |
| Source | *pseudo-F* | *p* | | | | |
| Species | 3.46 | < 0.001 | *T. mitchelli* | | a | |
| | | | *T. pearsei* | | a | b |
| | | | *T. dzilamensis* | | | b |
| ***T. mitchelli*** | | | | | | |
| Source | *pseudo-F* | *p* | | | | |
| Hanging control | 4.33 | < 0.001 | | | | |
| Salinity challenge | 0.39 | 0.884 | | | | |
| Time | 1.38 | 0.187 | | | | |
| Interaction | 1.23 | 0.286 | | | | |
| ***T. dzilamensis*** | | | | | | |
| Source | *pseudo-F* | *p* | | | | |
| Hanging control | 4.05 | < 0.01 | | | | |
| Salinity challenge | 1.03 | 0.391 | time 1 | a | | |
| Time | 3.32 | < 0.01 | time 2 | | b | |
| Interaction | 1.15 | 0.329 | time 3 | | | c |
| ***T. pearsei*** | | | | | | |
| Source | *pseudo-F* | *p* | | | | |
| Hanging control | 1.49 | 0.19 | | | | |
| Salinity challenge | 1.71 | 0.134 | | | | |
| Time | 1.38 | 0.182 | | | | |
| Interaction | 1.26 | 0.264 | | | | |

throughout all exposure times (Table 2). This was also reflected in data points from individuals in the moderate and extreme salinity challenge being comparatively interspersed (Fig 3B) Cellular damage, measured as LPO, had a larger impact in the PCoA3 (S2 Table and S1 Fig), and no patterns in this axis could be associated to salinity or time.

## T. pearsei

The PCoA applied to the antioxidant system and oxidative damage measured in *T. pearsei* showed no clear patterns of a conspicuous shift in the response to salinity (Fig 3C). Furthermore, no significant differences in the BI among control and challenged individuals could be found (Table 2). The overall large dispersion of the data could be responsible for the lack of statistical differences (Table 2 and Fig 3C and S1 Fig).

## T. dzilamensis

The ordination of the antioxidant system and oxidative damage in *T. dzilamensis* again showed that separation in the horizontal axis was mostly due to the contribution of SOD and GST (Fig 3D and S2 Table). However, control individuals had low concentrations of these antioxidant enzymes (were located at the left), whereas those subject to either salinity challenge had high

concentrations (were located at the right-hand side of the horizontal axis, Fig 3D and S1 Fig). Permutational MANOVA showed that BI in control individuals were significantly different from those of experimental treatments (Table 2), confirming a strong reaction from the anti-oxidant system as a response to changes in salinity. The time of exposure had a significant impact on this response, which was independent of the magnitude of the salinity challenge, indicating that changes in BI through time were similar for all individuals. The ordination shows that changes go from low SOD and GST, and high CAT activity in control individuals towards a reduced activity in CAT and greater amounts of PO, GST and SOD as exposure time increased (Fig 3D and S2 Table). Pair-wise comparisons among the time of exposure showed that enzyme activity and the oxidative damage confirmed that these differences in BI were statistically distinguishable between all sampling moments (Table 2).

## Lactate

Differences in lactate content among the three species in native salinity (controls) were not statistically significant (Table 3), suggesting that these three species have a similar lactate content during routine activity when they are not osmotically challenged.

**Table 3. Results of permutational ANOVAs on lactate content measured in *T. mitchelli*, *T. pearsei* and *T. dzilamensis* at three moments ($T_1$, $T_2$ and $T_3$) after being exposed to either a moderate or an extreme salinity challenge.** Hanging controls were measured in individuals of each species collected from their native salinity. See descriptions in the text for references to the time of exposure and water salinity used to challenge each species. The pseudo-*F* and p values for each source of variation is given together with number of unique permutations obtained for the corresponding test; different letters represent mean values that were statistically distinguishable ($p < 0.05$).

| General MANOVA | | | Groups of means following post-hoc tests | | |
|---|---|---|---|---|---|
| All species native salinity (controls) | | | | | |
| Source | *pseudo-F* | *p* | | | |
| Species | 2.35 | 0.114 | | | |
| *T. mitchelli* | | | | | |
| Source | *pseudo-F* | *p* | | | |
| Hanging control | 2.12 | 0.158 | | | |
| Salinity challenge | 8.04 | $< 0.05$ | moderate | a | |
| Time | 0.72 | 0.501 | extreme | | b |
| Interaction | 1.69 | 0.219 | | | |
| *T. pearsei* | | | | | |
| Source | *pseudo-F* | *p* | | | |
| Hanging control | 1.09 | 0.304 | moderate-time 1 | | |
| Salinity challenge | 15.07 | $< 0.001$ | moderate-time 2 | | |
| Time | 2.57 | 0.093 | moderate-time 3 | | |
| Interaction | 3.91 | $< 0.05$ | | | |
| | | | extreme -time 1 | a | |
| | | | extreme -time 2 | a | b |
| | | | extreme -time 3 | | b |
| *T. dzilamensis* | | | | | |
| Source | *pseudo-F* | *p* | | | |
| Hanging control | 0.99 | 0.355 | moderate-time 1 | a | |
| Salinity challenge | 47.49 | $< 0.001$ | moderate-time 2 | a | |
| Time | 42.04 | $< 0.001$ | moderate-time 3 | | b |
| Interaction | 16.29 | $< 0.001$ | | | |
| | | | extreme-time 1 | a | |
| | | | extreme -time 2 | a | b |
| | | | extreme -time 3 | | b |

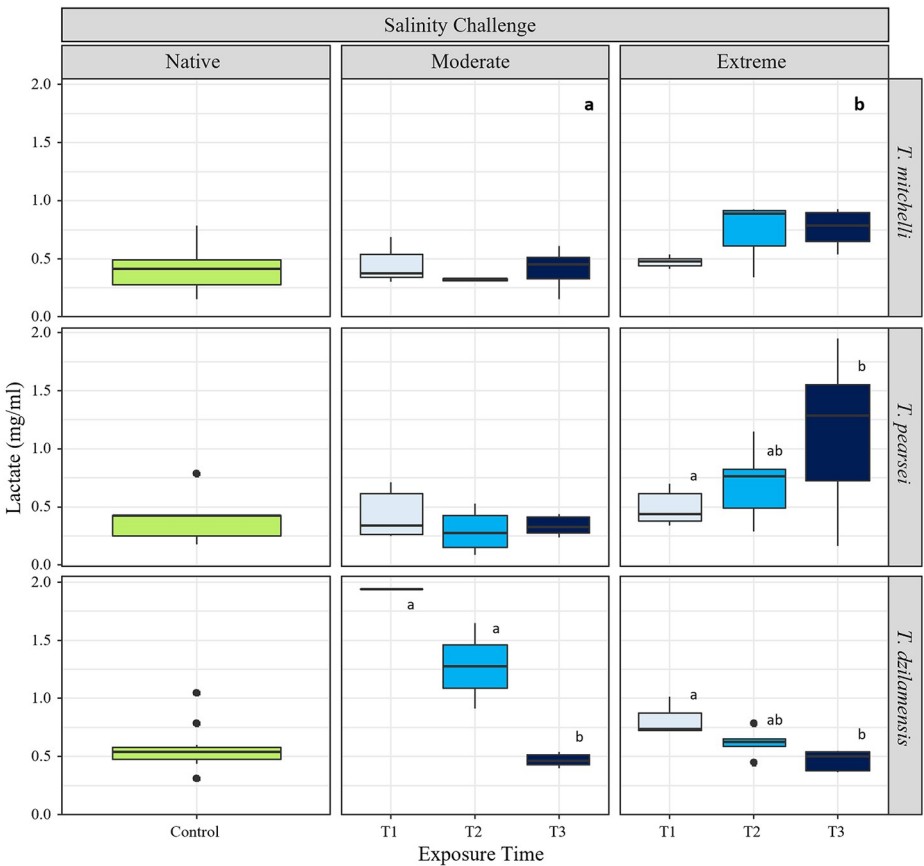

**Fig 4. Lactate content in *Typhlatya* species before and during salinity challenges.** Lactate content (mg/ml) in *T. mitchelli*, *T. dzilamensis* and *T. pearsei* under native salinity conditions (green boxplots) and after three moments (T$_1$, T$_2$ and T$_3$) of exposure to either a moderate or an extreme salinity challenge. Exposure times were 7, 15 and 80 minutes for *T. mitchelli*; and 15, 60 and 150 minutes for *T. pearsei* and *T. dzilamensis*. Native salinity represents salinity at collection site for each species (3 S$_p$ for *T. mitchelli*, 0.7 S$_p$ for *T. pearsei* and 34 S$_p$ for *T. dzilamensis*), moderate salinity trials were at 14 S$_p$, and extreme salinity trials were at 34 S$_p$ for *T. mitchelli* and *T. pearsei*, and 3 S$_p$ for *T. dzilamensis*. Bold letters represent significant statistical differences among salinity treatments, and letters at corresponding exposure times within a salinity show statistically significant differences.

Permutational ANOVAs on lactate content in *T. mitchelli* showed statistical differences related only to the magnitude of salinity change (Table 3), having higher content in the extreme salinity challenge (34 S$_p$) as compared to the moderate salinity challenge (Fig 4), regardless of the time of exposure (Table 3).

In *T. pearsei*, by contrast, differences in lactate were attributed by both the time and the magnitude of the salinity challenge (Table 3). Lactate was similar through exposure time in the moderate salinity, as there were no significant differences among times of exposure (Table 3). However, there was a gradual accumulation in lactate when exposed to extreme salinity (Table 3 and Fig 4).

In contrast to the other species, lactate in *T. dzilamensis* increased shortly after salinity transfer occurred and was subsequently reduced with exposure time (Fig 4). However, both the magnitude of the change and the time it took for levels to drop were different among the moderate and the extreme salinity challenges (Table 3). In the moderate salinity challenge T$_1$ and T$_2$ did not show statistically significant differences from each other, and both were statistically significantly different from T$_3$, suggesting there was a large increase in LAC immediately

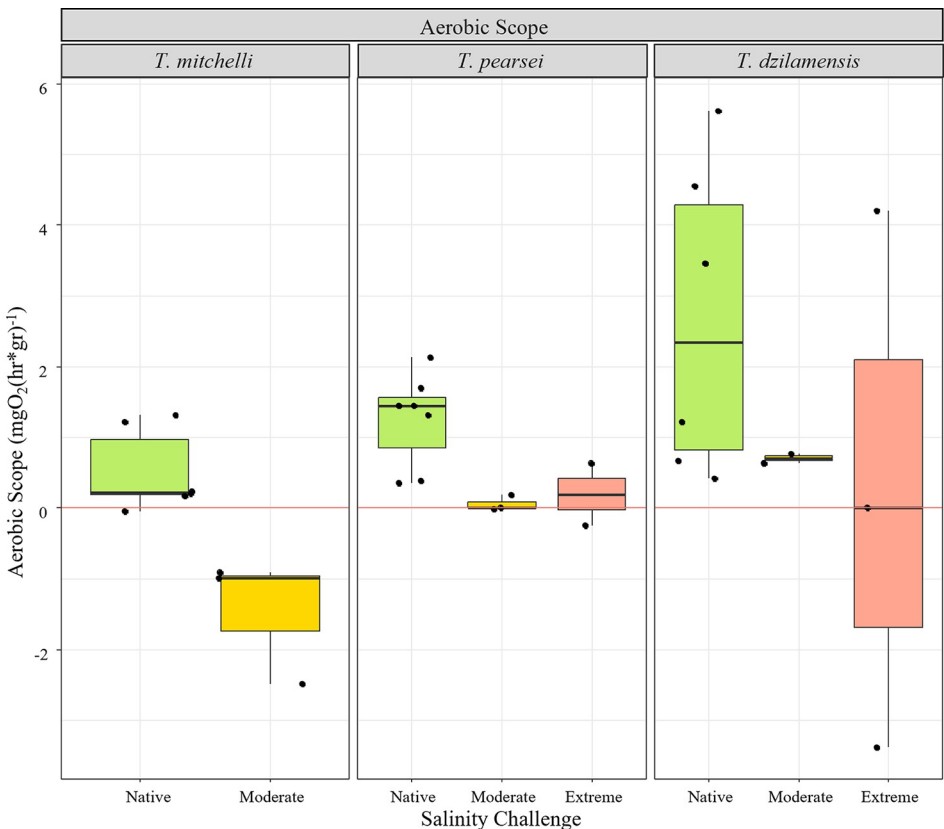

**Fig 5. Aerobic scope of *Typhlatya* species in native salinity and during moderate and extreme salinity challenges.**
Aerobic scope (mgO$_2$ h$^{-1}$ g$^{-1}$) in *T. mitchelli*, *T. pearsei* and *T. dzilamensis* under native salinity conditions and after 40 minutes of exposure to either a moderate or an extreme salinity challenge. Native salinity represents salinity at collection site for each species (3 S$_p$ for *T. mitchelli*, 0.7 S$_p$ for *T. pearsei* and 34 S$_p$ for *T. dzilamensis*), moderate salinity trials were at 14 S$_p$, and extreme salinity trials were at 34 S$_p$ for *T. mitchelli* and *T. pearsei*, and 3 S$_p$ for *T. dzilamensis*. Dots within each plot indicate individuals.

after salinity transfer which was not reduced until T$_3$ (150 min.). In the extreme salinity challenge lactate increase was smaller, and its reduction was gradual as only T1 and T3 were statistically significantly different (Table 3).

## Aerobic scope

In their native conditions, *T. mitchelli* had the lowest AS, followed by *T. pearsei*, and finally, *T. dzilamensis* had the greatest AS of all three species. However, *T. dzilamensis* was also the species with the widest variation with three individuals showing values between 3.5 and 6 mg O$_2$ l$^{-1}$ h$^{-1}$ g$^{-1}$, whereas the rest had values as low as 0.42 mg O$_2$ l$^{-1}$ h$^{-1}$ g$^{-1}$ (Fig 5).

All *Typhlatya* individuals showed a reduction in AS when exposed to a moderate salinity compared to their native salinity (Fig 5): *T. mitchelli* had a negative AS, indicating that 14 S$_p$ was enough to reduce the MMR after swimming to values below RMR at this salinity. The AS in *T. pearsei* was positive and near zero, suggesting the RMR has a similar (yet lower) magnitude than the MMR after 5 minutes of swimming and a total of 40 min at 14 S$_p$. When challenged with 14 S$_p$, *T. dzilamensis* also reduced its AS, both its central tendency and its variation to values below (yet positive) their AS in native conditions.

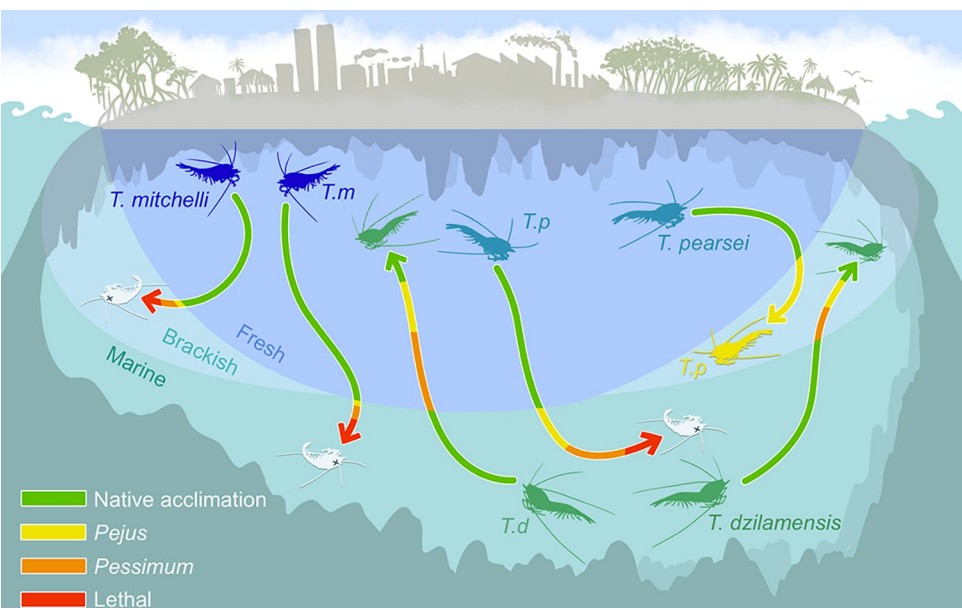

**Fig 6. Conceptual model of the physiological stress of acute salinity changes in *Typhlatya* species.** *T. mitchelli* shows extreme physiological stress after salinity changes to both brackish or marine groundwater, *T. pearsei* showed moderate stress indicators in the brackish trials and extreme stress in the marine groundwater trials, *T. dzilamensis* shows extreme stress immediately after salinity change, but stress is reduced with time suggesting this species can tolerate acute changes in salinity. Native salinity for *T. mitchelli* (Tm) and *T. pearsei* (Tp) is the fresh groundwater layer (top), as the marine groundwater layer (bottom) is for *T. dzilamensis*. The intermediate salinity challenge is depicted as the brackish groundwater layer at the left and right sides of the groundwater in this conceptual model. Stress range categories are shown as processes that shift from the native acclimation (green) to *pejus* (yellow), *pessimum* (orange) or *lethal* (red) after traversing the theoretical haloclines.

When subject to the extreme salinity challenge, only two individuals of *T. pearsei* had consistent oxygen measurements which could be used, one with a negative and one with a positive AS value. Something similar occurred with *T. dzilamensis*, where the oxygen consumption of three individuals showed a large dispersion over both positive and negative AS values (Fig 5).

## Discussion

In this study, we observed differences in the physiological response to salinity of the three *Typhlatya* species over a short period of time, measured through their metabolic rates (Figs 2 and 5), changes in their antioxidant system (Fig 3) and lactate accumulation (Fig 4). Results obtained here indicate for the first time that *T. mitchelli* is physiologically restricted to the FG environment of the anchialine caves (Fig 6), showing a high metabolic rate after salinity changes followed by a total collapse. In contrast, *T. dzilamensis*, which is the only species observed in the MG environment, showed a metabolic and physiological response that suggests it is capable of transitioning through the haloclines into both brackish and FG (Fig 6). An intermediate condition was observed in *T. pearsei*, which showed a positive physiological response in the intermediate salinity challenge (brackish groundwater) but could not endure 150 minutes of the extreme salinity challenge (Fig 6).

Similar RMR across these species in their native salinity implies that these species routinely invest similar amounts of energy, despite being on opposite sides of the halocline. Salinity transfers from native to target salinities showed an immediate increase in MR in all individuals, indicating that such acute salinity changes demand an increase in energy production for all species. Initially, such increase in RMR could be expected because active osmoregulation

requires immediate energy allocation [72–74]. However, changes in RMR alone are not enough to interpret the response as beneficial or detrimental, suggesting that both the time of exposure and a suit of other indicators are necessary to untangle the physiological response.

Salinization of groundwater due to over extraction of freshwater is a reality in some regions, where an increase in salinity has modified the community structure of stygobionts and reduced its biodiversity [75]. Furthermore, combined effects of salinity stress with other abiotic factors may have additive detrimental effects on populations [13, 76], leading to the migration of populations (if habitat connection allows) or local extinction. However, considering that osmoregulatory mechanisms in crustaceans may vary among species that co-occur in the same ecosystem, regardless of phylogenetic or ecological ancestry [74], species specific responses to salinity variations may be expected [74, 75, 77], which makes regional studies and targeting groundwater-restricted species urgent. This work adds another case study where closely related species which are restricted to the same ecosystem, have differential salinity tolerance and, thus are affected in different degree by salinity changes [72–74].

Phylogenetically, the *Typhlatya-Stygiocaris-Typhlopatsa* complex (TST) has a marine and Tethyan origin [18, 78], yet ten out of the 17 lineages inhabit low salinity (<5 psu) environments [26]. In particular, the *Typhlatya* pertaining to the Yucatan + Cuba clade were found to have a low salinity most recent common ancestor [26]. However, *T. dzilamensis* and occasionally *T. pearsei* are the only species of the clade that have been observed in MG [16, 17, 28, 29, 79]. This suggests that *T. dzilamensis* diverged from a FG ancestor and colonized the MG as a novel trait [26], while *T. pearsei* and *T. mitchelli* conserve the ancestral FG habitat. In this regard, Havird et al. [74], propose that freshwater species would have to increase the intracellular pool of osmolytes to keep cellular volume constant, an additional restriction for the colonization of higher salinities which could be taxing for FG species. This could imply that *T. dzilamensis* has maintained some ancestral characteristics that enable it to "return" to the FG environment. In previous study on the tolerance of salinity changes in anchialine shrimp *Halocaridina rubra* found that tolerance to salinity could be linked to a defensive priming strategy [74, 80, 81], a feature that has also been suggested for *T. dzilamensis* [17] and is apparently not shared by *T. mitchelli* or *T. pearsei*.

All physiological responses to stress require energy, and the allocation of energy to essential and non-essential processes can be shifted according to the degree of stress (for a review, see Sokolova [38]). Therefore, the effect an environmental change has on the metabolic traits reflects the stress of each organism in the new environment [36, 38]. Following the ideas of energy-limited tolerance to stress and the oxygen and capacity-limited thermal tolerance, the physiological indicators obtained for all species during our experiments may be categorized into a *pejus*, *pessimum*, or *lethal* stress range [35, 38, 40, 42, 82]. However, the question of how long individuals may survive under diverse stress levels, or if they would eventually overcome the environmental variations by means of acclimation (*i.e.* by reducing the cost of maintenance) cannot be fully answered at this point and requires addressing each species one by one.

The moderate and extreme salinity trials in *T. mitchelli* had such an impact that, despite the five-fold increase in MR increased, these individuals could not swim and displayed tail flips, suggesting that any surplus energy may eventually be invested in escaping. Furthermore, lactate accumulation increased in the extreme salinity challenge after 15 and 80 minutes, indicating anaerobic metabolism was activated to add to the energy budget. The AOS was completely modified from the native configuration implying that ROS mitigation occurred, and the negative AS is indicative that all available energy was likely allocated into a protection strategy, in an attempt to avoid the osmotic shock of and an imminent physiological collapse. These features together with the escape behaviour observed in *T. mitchelli* show that such acute transitions to 14 $S_p$ or 35 $S_p$ represent a

stress, consistent with a *pessimum* and *lethal* salinity ranges. Consequently, it is possible to conclude that *T. mitchelli* is a stenohaline species restricted to the FG habitats of the YP.

After both moderate and extreme salinity changes, *T. pearsei* showed lower increments in MR than observed in the other species. The increase of MR alone could suggest two opposing explanations, either i) this species lacked the energy surplus required to meet the energy demands of salinity change; or ii) the mechanisms involved in osmoregulation and maintaining homeostasis are highly efficient. Considering the low AS following the abrupt salinity change, it becomes clear that *T. pearsei* was operating near its MMR, leaving a narrow energy margin for processes other than homeostasis maintenance.

The lethal outcome of the extreme salinity challenge for *T. pearsei* (with 1 of 6 surviving individuals) highlights the importance of exposure time in for evaluating its tolerance. *T. pearsei* individuals which initially showed a positive physiological and behavioral response were overwhelmed after 150 minutes, indicating that *T. pearsei* would not survive long-term exposures after traversing through a halocline into MG. However, all *T. pearsei* individuals survived 150 minutes in brackish groundwater and short-term exposures to MG, which could be ecologically advantageous for the species.

Additionally, lactate contents during the moderate salinity challenge remained relatively constant in *T. pearsei*, implying that the energy required was obtained solely from aerobic mechanisms, which is consistent with a *pejus* stress range [38, 83]. It is also noteworthy that the AOS showed no significant changes both in moderate and extreme salinities, consistent with the low increase in MR. This suggests that a low metabolic rate might be an energetic strategy for *T. pearsei*, possibly due to the oligotrophic nature of the environment.

Despite the limited information on the osmoregulatory capacity of this species, the exposure of *T. pearsei* to moderate salinity (14 $S_p$) for 150 minutes suggests a high efficiency in osmoregulation, with a demand from anaerobic metabolism only required under the extreme condition (34 $S_p$) after150 min. The lactate accumulation observed during the extreme salinity challenge indicates a stress-induced transition to anaerobiosis, suggesting that the extreme salinity challenge represents a borderline limit to their tolerance, which is consistent with the *pessimum* and *lethal* stress ranges, and could only be tolerated for a short period of time [38, 83]. This indicates that *T. pearsei* can tolerate MG environments only for a limited time.

Our results are unable to predict how long *T. pearsei* could have endured the moderate salinity scenario; however, the observed low AS may have fatal consequences for the long-term survival and fitness of the population [38], unless they can acclimate to the new conditions over time and return to an optimal physiological condition [38, 83].

*T. dzilamensis* showed an increase in MR immediately after the abrupt salinity transfer, followed by a reduction in MR with increasing exposure time. Unlike the other species, they maintained a positive AS during salinity trials and did not exhibit signs of stress-related behaviour. Additionally, and unlike the other species, these individuals showed an initial increase in lactate accumulation immediately after the salinity transfer, which subsequently decreased as exposure time passed. These indicators suggest that during the salinity trials, *T. dzilamensis* individuals were initially energetically challenged as they resorted to partial anaerobiosis, which is consistent with a *pessimum* stress range. However, their physiological response compensated for the initial salinity shock, reducing lactate accumulation after a short period of time (past T1; *i.e.* 60 minutes) and yielding a low, yet positive AS. Furthermore, ther AOS changed gradually as the exposure time increased. Unlike *T. mitchelli*, the AOS in *T. dzilamensis* showed high activity of SOD, and GST, and a low accumulation of LPO and PO, indicating that the AOS was efficient in mitigating ROS damage during the salinity trials.

Haloclines in anchialine environments are dynamic in depth and magnitude of salinity difference, especially in conduits closer to the coast [12, 14, 15]. As the FG lens decreases during

the dry season and water runoff to the coast is reduced, the halocline tends to decrease in depth gradually, and to change from FG to MG without an intermediate brackish layer [14]. When the rainy season recharges the FG lens, and flow to the coast becomes stronger, or due to storms that rapidly recharge the aquifer, salinity may drastically change in depth and extension causing long lasting effects in the groundwater habitat [15]. Such environmental changes have a significant impact on habitat availability and physiological stress, frequently resulting in fatal consequences for groundwater communities [12]. Thus, environmental changes in salinity may determine the distribution of species within groundwater environments [24, 84–86].

## Conclusions

Our results indicate that the fundamental niche of *T. mitchelli* is restricted to the freshwater portion of the anchialine environments, and they would not endure such salinity variations. However, our results are not conclusive on the tolerance of *T. pearsei* to brackish water, it is possible that if the transition to brackish salinity is gradual, or after a longer acclimation time, they may reduce the metabolic demand of the initial responses and eventually have a greater AS, granting *T. pearsei* a fundamental niche of fresh and brackish groundwater. We show that *T. pearsei* is unable to endure acute salinity transfers into fully MG for more than 150 minutes, but is functional for short periods of time, which could be an advantage for escaping freshwater stenohaline predators or could determine the population's survival under the dynamic halocline scenarios. Ecologically, the capability *T. dzilamensis* individuals to resist acute salinity changes from MG into brackish and FG is interpreted as the ability to cross the haloclines, which would enable them to obtain nutrients from both sides of the haloclines and explain the difference in carbon uptake from the other species [16].

In this work, we found compelling evidence that the change in salinity alone (either due to global warming or anthropogenic influence) may have catastrophic consequences for the majority of the *Typhlatya* species inhabiting the YP. Additionally, the additive impact of several changing factors (OM input, temperature, oxygen, etc.) would be expected to have an exponential negative impact on the survival of cave communities [5, 13, 38, 76]. Therefore, studying the impacts of multiple stressors, including anthropic waste (heavy metals, pesticides, and excess organic input) is urgent to identify priorities in groundwater management, as the role of groundwater communities as ecosystem service providers is still a developing field, and the impact of losing stygobiont species could have direct impacts on the water quality [87, 88].

Our results demonstrate that, although these species are closely related, their different physiological capacities have a great impact on their ecological niche and their tolerance to the changing environment. This also adds to the understanding of groundwater communities and could be the basis for upcoming works in conservation physiology for these ecosystems.

Finally, this work shows that coastal populations of *T. mitchelli* are at greater risk of habitat reduction with the upcoming climate change scenarios than populations of *T. pearsei* and *T. dzilamensis*.

## Supporting information

**S1 Video. *Typhlatya*'s reactions to a halocline.** The first segment of the video shows a *T. mitchelli* as it swims down into the fresh groundwater (FG) portion of the cave, reaches the halocline, and uses a swift tail-flip to retrieve from the marine groundwater (MG). This behavior is repeated every time it reaches the halocline. The second segment shows a *T. dzilamensis* as it swims from the FG into the MG and remains in the latter without any reaction to the salinity change. Haloclines are visually recognized by either the refraction of light (first segment) or by

a blurry mix of water layers (second segment).
(AVI)

**S1 Table. Eigenvalues and eigenvectors that resulted of the principal coordinate analysis (PCoA) applied on six indicators of the antioxidative system and oxidative damage measured in *T. mitchelli*, *T. pearsei* and *T. dzilamensis* at their native salinities.** The amount of explained variation by each principal coordinate is expressed in absolute, relative and cumulative magnitudes. Antioxidative enzymes were catalase (CAT), glutathione S-transferase (GST), total glutathione (GSH), and superoxide dismutase (SOD). Oxidative damage was quantified through protein carbonylation (PO) and lipid peroxidation (LPO).
(TIF)

**S2 Table. Eigenvalues and eigenvectors of the principal coordinate analysis (PCoA) applied on six indicators of the antioxidative system and oxidative damage measured in *T. mitchelli*, *T. pearsei* and *T. dzilamensis* at their native salinities and at three moments ($t_1$, $t_2$ and $t_3$) after being exposed to either an intermediate or a severe salinity challenge.** The amount of explained variation by each principal coordinate is expressed in absolute, relative and cumulative magnitudes. Antioxidative enzymes were catalase (CAT), glutathione S-transferase (GST), total glutathione (GSH), and superoxide dismutase (SOD). Oxidative damage was quantified through protein carbonylation (PO) and lipid peroxidation (LPO).
(TIF)

**S1 Fig. Antioxidant system enzyme activity and cellular damage indicators through time in *Typhlatya* species in native salinity (green), and after an acute salinity change into moderate (yellow) or extreme salinity.**
(TIF)

## Acknowledgments

The authors gratefully acknowledge Nelly Tremblay for her assistance with data acquisition, Ariadna Sanchez Arteaga for her laboratory assistance, Quetzalli Hernandez, Dorottya Angyal, Michel Vázquez and Arturo Mora for their cave diving and field support, Parque Ecológico Chikin Ha and the cenote Nohmozon for granting permission to conduct the field experiments. Fieldwork was supported by Dr. Nuno Simoes by providing specialized equipment. The authors are grateful for the thorough revision, comments and suggestions of two anonymous reviewers which have greatly improved this manuscript.

## Author Contributions

**Conceptualization:** Efrain M. Chávez Solís, Maite Mascaro, Carlos Rosas.

**Data curation:** Efrain M. Chávez Solís, Maite Mascaro, Carlos Rosas, Claudia Caamal Monsreal.

**Formal analysis:** Efrain M. Chávez Solís, Maite Mascaro.

**Funding acquisition:** Carlos Rosas, Gabriela Rodríguez-Fuentes.

**Investigation:** Efrain M. Chávez Solís.

**Methodology:** Efrain M. Chávez Solís, Maite Mascaro, Carlos Rosas, Gabriela Rodríguez-Fuentes, Claudia Caamal Monsreal, Kurt Paschke, Fernando Díaz, Denisse Re Araujo.

**Resources:** Maite Mascaro, Carlos Rosas, Gabriela Rodríguez-Fuentes.

**Supervision:** Maite Mascaro, Carlos Rosas, Gabriela Rodríguez-Fuentes, Kurt Paschke, Fernando Díaz.

**Validation:** Maite Mascaro, Carlos Rosas, Gabriela Rodríguez-Fuentes, Denisse Re Araujo.

**Writing – original draft:** Efrain M. Chávez Solís.

**Writing – review & editing:** Efrain M. Chávez Solís, Maite Mascaro, Carlos Rosas, Gabriela Rodríguez-Fuentes, Claudia Caamal Monsreal, Kurt Paschke, Fernando Díaz, Denisse Re Araujo.

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
