## [Decision Letter · Decision Letter 0]

8 Dec 2023

PONE-D-23-27494Are haloclines distributional barriers in anchialine ecosystems? Physiological response of cave shrimps to salinity.PLOS ONE

Dear Dr. Chávez Solís,

Thank you for submitting your manuscript to PLOS ONE. After careful consideration, we feel that it has merit but does not fully meet PLOS ONE’s publication criteria as it currently stands. Therefore, we invite you to submit a revised version of the manuscript that addresses the points raised during the review process.

We look forward to receiving your revised manuscript.

Kind regards,

Sanja Puljas

Academic Editor

PLOS ONE

Journal Requirements:

3. Please expand the acronym “CONACYT” (as indicated in your financial disclosure) so that it states the name of your funders in full.

**Additional Editor Comments:**

Reviewers' comments:

Reviewer's Responses to Questions

**Comments to the Author**

1. Is the manuscript technically sound, and do the data support the conclusions?

Reviewer #1: Yes

Reviewer #2: Yes

2. Has the statistical analysis been performed appropriately and rigorously? 

Reviewer #1: Yes

Reviewer #2: N/A

3. Have the authors made all data underlying the findings in their manuscript fully available?

Reviewer #1: No

Reviewer #2: Yes

4. Is the manuscript presented in an intelligible fashion and written in standard English?

Reviewer #1: Yes

Reviewer #2: Yes

5. Review Comments to the Author

Reviewer #1: In the current manuscript, the authors explore salinity tolerance and salinity-induced stress responses in 3 species of anchialine shrimps from the Yucatan peninsula. They use metabolic measures and enzyme activities to show that two of the three species likely have minimal tolerance to changing salinities and are confined to the freshwater portion of the habitats, while the third species is more tolerant and may be able to traverse the halocline. Overall, I found the manuscript interesting and a welcome addition to physiological work on anchialine organisms. Little is known about these species and examining salinity responses is a good first step in characterizing how they have adapted to these habitats. I think PLoS One is also an appropriate outlet for this work. However, I do have several criticisms that the authors should address during revision.

1) More detail needs to be given about acclimation times during the metabolic rate experiments. After shrimp were collected from the habitats and brought up, how long were they allowed to adjust to their new settings before metabolic rate measurements began at the different salinities? In most experiments of this kind, animals are allowed to settle in for a period of time before measurements begin (or before the data are recorded), because handling stress can cause MR to increase. In all the species, MRs are high initially at test salinities and then decrease over time. The authors interpret this as a physiological response to salinity, but it may be a result of handling stress during setting up the experiment and/or transfer of animals to the new salinity. If shrimp were brought up and then placed in the native salinity for control measurements over 150 minutes, the authors could address this by showing the same time series data (over 150 mins) for the control vs. test salinities to see if MRs were initially elevated there as well before lowering. Based on Fig 1, it seems this might not be the case, so the authors may not be able to disentangle time from salinity. In any case, stress due to handling and initial responses to the experimental apparatus (not salinity) should be discussed. The samples sizes for each measurement/experiment and species should also be referenced in the main text, not just Fig 1.

2) In interpreting the results, the authors should consider direction of salinity challenge as a possible explanation. T. mitchelli and T. pearsei were undergoing freshwater to seawater transfers and had increased stress/low survival. T. dzil. underwent an opposite transfer and had high survivorship. Given that the atyids are a freshwater family, increased salinity might be a natural stressor, while T. dzil. has switched to being a seawater specialist, but still maintains the ability to survive in freshwater. In truly euryhaline species, researchers often acclimate all species to the same salinity and then undergo a transfer in the same direction. While that isn’t possible in this system, it is worth noting that direction of transfer also covaried with tolerance here. It should also be noted that because of the nature of the system, an entirely controlled and comparable experiment is not possible.

3) I found the presentation of the results somewhat confusing. This is especially true for figures 3 and 5. For fig 3 the main issue is that the PCA is difficult to interpret given the overlap of the points in many panels. Adding a mean and ellipse for each treatment would go a long way towards letting the reader see how the treatments are affecting the PCA values. I’d also suggest adding a set of supplementary figures similar to Fig2 and 4 for each of the enzymes examined, so readers interested in a particular enzyme can see how its activity changed with salinity in each species. Similarly, in Fig 5 the interesting comparison is how AS changes with salinity within each species, not differences between species at a particular salinity. I’d suggest reformatting this figure to match the layout in Figs. 2 and 4.

4) Throughout the manuscript there are issues with writing clarity. In general these are not major flaws and the intent is clear (I’ve outlined some of these below), but a thorough reading for syntax and grammar would be appreciated.

Minor comments

1) Line 23 – “by the haloclines” should be “by haloclines”

2) Line 32 – “novel physiological characteristics” is too vague. I suggest removing this and emphasizing that some species may be euryhaline while others are restricted to the freshwater portion of the habitat.

3) Line 52 – remove “the” before “rapidly advancing climate change”

4) Line 58 – change “is” to “are” and remove “the”

5) Line 113 – change “resource” to “resort”

6) Line 136 – list collection parameters for T. pearsei here as well.

7) Line 190 – change “trough” to “through”

8) Line 225- change “essay” to “assay”

9) Line 309 – what statistics were used to analyze the aerobic scope data? I didn’t see this in the statistics section

10) Line 319 – the laying out of all the different comparisons is quite confusing. I might suggest condensing this section and having a few lines for each species stating the general trend and then maybe the significant differences.

11) Prior to getting into the MR data, I’d suggest a few lines outlining the behavior and survival of each species instead of listing this later. This will give the reader an idea of general stress responses before getting into the MR data. In general, I think its ok to have a paragraph for each species where the behavior/survival observations are noted first, then the MR trends and significant differences. It is also difficult to keep track of the species if one isn’t familiar with them, so it is useful t reiterate for each species where they were collected (FW or SW) and which direction they were transferred in.

12) Line 563 – this is a good point that I would like to see expanded upon – increased MR could be interpreted as an adaptive/beneficial response (they are able to mount a physiological response) or a maladaptive response (they are burning lots of energy). Because these interpretations are polar oppsites, its important to interpret the data here in both ways and discuss what either interpretation could mean for stress, etc.

13) Line 615: “which can”?

Reviewer #2: Dear authors! I have reviewed your manuscript: “Are haloclines distributional barriers in anchialine ecosystems? Physiological response of cave shrimps to salinity.” It is a relevant and timely study bringing novel data that highlight potential issues certain species might experience in the rapidly changing world due to changes in salinity in a sensitive and rare environment. I did not find any major issues regarding the content but think that the paper is poorly organised; not clearly written and needs a major makeover of many parts. Please see my comments and suggestion below on how to improve this.

INTRODUCTION

I think the introduction needs a thorough makeover. A substantial part of text (3rd, 4th, and 5th paragraph) is now very technical, devoted to introducing common, well-known physiological traits, their proxies and how to measure these. This has nothing to do with the story you are telling but is a much more related to the methods you used to assess sensitivity to salinity. I recommend you move most of this text to the appropriate parts of the methods section. In the intro, possibly just in the last paragraph, mention just the essentials. This will free-up space which I suggest you use to build a more solid story. I think you should make climate change (and anthropic) driven changes in environment and their effect on species living along haloclines and potential conservation issues as your main thread and strengthen it. Please include relevant recent review studies on threats to cave life and habitats like Mammola (many papers), Wynne et al. 2021, but also one of the landmark papers on anchialine cave ecology (Sket, 1996). Also, more specifically introduce what is know of salinity sensitivity/tolerance in cave animals in general. Due to increased threats of anthropic salination of freshwater, recently a few relevant studies were published (e.g., Castaño-Sánchez et al 2020, Jemec et al. 2022) and I suggest you refer to them either in introduction or devote a paragraph for this in discussion. Finally, please improve the presentation of your model system in the 6th paragraph. Say why are these shrimps a good model for your question, and let the reader know the essentials of the model system required to understand this.

Sket, B 1996: https://doi.org/10.1016/0169-5347(96)20031-X

Mammola et al: https://doi.org/10.1111/brv.12642;
https://doi.org/10.1111/brv.12851;
https://doi.org/10.1093/biosci/biz064;
https://doi.org/10.1177/2053019619851594

Wynne et al. 2021: https://doi.org/10.1111/conl.12834

Castaño-Sánchez et al 2020: doi: 10.1038/s41598-020-69050-7

Jemec et al. 2022: https://doi.org/10.1016/j.ecoenv.2022.113456

First lines: support your claims with references

Lines 54-57: first paragraph of the intro should introduce the broad topic; this interrupts the flow, please remove and reuse maybe in the last parts of the introduction; for example the 6th paragraph

56-57: No need for abbreviations, in my opinion. Just use the full phrasing throughout the manuscript.

58-60: flip the order of the sentence

63: FG and MG, usually your abbreviate as FGW and MGW

68-69: impossible to understand without checking Box 1, but actually also not explained in the box, but in the 3rd paragraph of intro. Please rewrite so that the question is instantly understandable.

71: no need for a capital letter in Oxygen

78: a fullstop is missing after references

85: start a new sentence after “decreases”

86: rephrase “have exceeded”

90 & 95: lethal in italic

LINE 99: comma after demand

Line 105: no need to use capital letters

Line 121: Please reconsider expressing your units of salinity (UPS) in the new standard, TEOS-10, so in promilles, g/kg, also dimensionless.

121-122: delete the sentence. It only adds confusion. Not clear what is this Yucatan-Cuba clade. Are Typhlatya part of it or not.

124: not clear what SGW is. Did you mean MGW?

126-130: very long sentence and thus hard to comprehend. Consider breaking into two or three shorter sentences.

131: and should not be written in italic

132: not clear what you mean by “compared such consequences”. Rephrase.

132: change metabolic rate to routine metabolic rate, otherwise the abbreviation does not match

METHODS

Animal collection

It is not clear where T.pearsei was collected: in the solely freshwater cenotes? It is also not clear whether T. michelli was collected also from these two, or just from the Pandrosa system. I guess T.dzilamensis was collected only in the last system. From the next subchapter it seems you did not sample in the Tza Itza cenote at all. Why do you mention it here?

Already in the introduction, you could say and explain more about the distribution of these shrimps and if and how often they co-occur. This relates to my previous comment in intro on improving model system description.

Also, I suggest you add a sentence to say how far apart are these cenotes, are they connected somehow, and so on. So improve the description of geography, collecting sites.

You should say the total number of animals per species. Was it time consuming to catch them? Are these numbers a significant proportion of the populations? Add this info so the reader can get a feeling for the effort it takes just to collect animals in this difficult habitat, and can better appreciate your work.

Line 139: capital letter for system

After animal collection you could open a new subchapter in which you could explain the choice of your physiological traits and proxies, that I suggested to move from the introduction into the methods.

Next subchapter would be “experimental setup” where you describe the materials you used in your experimental protocols. Now this is included mostly in Experiment 1.

Salinity scenarios

Please explain how you measured salinity at collection sites. Did you measure also throughout the vertical column to reconstruct the halocline?

Line 150: a space is missing before the references.

Line 158-159: change UPS to PSU

Experiment 1

Altogether not clearly described what was done. Very hard to follow. Exclude the experimental setup and focus on the protocol.

You should clearly say the sample sizes per species for this part of the experiment and also the subsequent ones. This is not clear enough. Also, not clear whether between the three experiments individuals were reused or a specific individual was used only in one experiment. Please explain.

Line 172: change MR to RMR

Line 180: change metabolic rate (MR) to routine metabolic rate (RMR)

Line 186 & 188: replace weight by mass, these are two different physical quantities and you measured mass, not weight

Line 189 & 190 & 216 & 235: add space between number and related unit

Line 205: replace Nitrogen by nitrogen

Experiment 3

It is not clear how the control treatment was performed. Please add a short explanation.

I guess you had to weight individuals also for the previous two experiments, but you do not mention it. Add.

Statistical analyses

Delete the first sentence and move it to the very end of statistical methods. Say that plots and analyses were done in R. The latter is missing now. Please also include and cite the R packages used as well as functions with which you performed the main model and post-hoc tests.

Either add subtitles or start each paragraph so, that it is instantly clear to which of the above experiments and data the analyses relate. Now it is not.

First paragraph (lines 266-279): I suggest reorganisation of text to improve clarity. First say that you tested for differences in RMR at native salinity between species and how. Second describe testing differences between salinity challenge and time. When describing the permutation ANOVA model please first say what was the dependent variable and what were independent variables, only then move to specifics of the permutational ANOVA. For general understanding it is more important to say you performed a two-way ANOVA model. For factor time use categories in min instead of t1, t2, t3. Lastly, describe how you performed post-hoc tests. Use main effects instead of model terms. You should adjust your p-values for multiple comparisons via one of the many available methods. Did you do it and just forgot to mention it?

Please explain ion text why you had to use permutational ANOVA and not normal ANOVA. Why did you not include an interaction between the two factors in your models? Also, did you consider performing one permutation ANOVA with an additional third factor “species”? Your models do not allow testing difference between species, which would be the most interesting thing to do in my opinion. The same questions and suggestions apply also to similar model procedures you describe next.

Second paragraph (lines 280-287): 1) Describe the procedure for baseline (control) data. 2) Describe the procedure for treatment data per species.

Line 280: usually abbreviated as PCA not PCoA.

Third paragraph (lines 288-301): It is not instantly clear the here you describe the analyses of PCA scores obtained from data on biochemical indicators. Please improve the text with this in mind. If you join this paragraph with the previous one, would already be helpful. Also, and critical, the MANOVA is insufficiently described. What were the dependent variables? How many PCA dimensions/axes did you include? Again, first describe the analyses of baseline data, controls, then move on.

Fourth paragraph (lines 302-309): Is very confusing and in my opinion mistakes are present. It is not clear what are “the same statistical procedures”. Especially as in the previous paragraph you describe MANOVA, a multivariate model, but here LAC is a single response variable and cannot be analysed with a multivariate model.

The description of the analyses related to aerobic scope are completely lacking. Please add.

RESULTS

General: keep in mind that your sample sizes per treatments are very low. (I understand that it is hard to obtain high numbers of these species.) Some non-significant results could be related to this. So, be reserved in your interpretations, and try not to over-interpret results. Use careful wording.

You mostly show the transformed data. I think it would be valuable to report also absolute values of RMR and other response variables. This would be useful for comparisons with results of other similar studies, or even metaanalyses. If possible please add summary values in main text, and/or raw values in supplement as a data file. Sorry, if this was done, but it is not clear as you do not give a reference where raw data are deposited.

Metabolic rates and behavior during salinity challenges

Line 312: add “salinity”; use past tense, not present tense when describing results: were not significant.

Line 318: I see here that you did include the interaction of factors time and salinity in your models. But you did not say this in the methods. Please add.

Lines: 319-321: These are the results of post-hoc pseudo-t tests. Please add the pseudo-t value along with the p-value.

Line 331-332: Judging from the plot it is not obvious that RMR is higher in moderate than in extreme salinity; it seems similar. Please recheck the data, plot and stats for mistakes. Also, the interaction was not significant, so your statement “time has a differential impact on the RMR regardless of the salinity challenge” is wrong. It would be true if the interaction would be significant.

Lines: 332-335: Here you describe the results of post-hoc test, but it is very hard to understand, and it is not done in the same way as for the previous species (stats are not included). I suggest you gather all post-hoc test results in a table. Then you can exclude the boring stats from the text and just summarize the main results/pattern. This way would be much more useful and improve clarity. On the other hand, current Table 1 in mostly redundant and can be deleted; see my comment below.

Table 1: There is no need to include the whole table; there is too much information that are not very useful for the readers. I would report only the pseudo-F and p-values. You can do this in text, there is no need for a table. For example: “RMR in native salinity conditions were not significantly different among species (pseudo-F: 0.09, p-value: 0.931).” I would advise to do similar for per species results (report in text model results for both factors’ main effects and their interaction). Minor: explanation of the table content is usually given above, not below the table. Below we only give minor annotations.

Figure 2: I recommend marking the significant differences with asterisks or letter codes on plots for moderate and extreme salinity challenge.

The last three paragraphs are confusing and need rewriting. You start with reports on behaviour, but it turns out you speak of RMR most of the time. This is confusing for the reader. I suggest doing the following: 1) keep only stuff related to behaviour and mortality and organize it into one paragraph; let it be the first one in this results section. 2) reuse the remaining text (lines 345-348, 352-356, and 360-364) and meaningfully include it in the description of RMR results.

Antioxydant system

It is hard to evaluate these results as the related statistical methods are confusingly and insufficiently described.

Line: 385: statistically significant differences

Line 388: statistically similar is an inappropriate term. Please use not statistically significantly different instead.

Line 392: replace sample by individuals or species. (2x)

Line 395: delete vertical.

Table 2: same comment and suggestion as for current Table 1.

Line 417: until here T. dzilamensis was always discussed after T. pearsei. Please keep this order throughout the manuscript.

Lactate

It is hard to evaluate these results as the related statistical methods are confusingly and insufficiently described.

Line 461: “in their” seems out of place. Please check.

Table 3: same comment and suggestion as for current Table 1 and 2 and comment that relate to that. Currently, reposting of results especially stats in this section is not clear and appropriate. Applying my suggestion will solve this issue.

Figure 4: replace “behavior” in the title with a more appropriate word like “content”

Aerobic scope

It is hard to evaluate these results as the related statistical methods are not provided in the methods section.

DISCUSSION

Figure 6: I like this graphic summary of your results very much!

Line 615: “which can” is out of place. Either a sentence is lacking, or you should delete these words.

Line 609-630: Nice point! I agree the dynamics of haloclines in the environment are an important selective factor. I am wandering if you have seasonal data regarding this for the cenotes where these shrimps live and where you sampled. Is the halocline at Tm location more stabile than at locations of the other two species? Can you add a few short sentences regarding this to this paragraph?

Your results implicitly suggest that colonisation of freshwater by marine species is easier than vice versa. This is interesting from an evolutionary standpoint. Does it match with existing knowledge, literature? Can you add a short discussion on this?

At the end of the discussion clearly connect back to the introduction, its main story on climate change impact in caves.

CONCLUSIONS

I think this chapter is not really needed. Here you are repeating what you said already a few times in the manuscript and at this point it sounds boring already. I suggest to drop out this section and include its novel parts (Tm being at greater risk due to rising sea levels) into the discussion.

6. PLOS authors have the option to publish the peer review history of their article (what does this mean?). If published, this will include your full peer review and any attached files.

Reviewer #1: No

Reviewer #2: No

---

## [Author Response · Author response to Decision Letter 0]

5 Jun 2024

Reviewer #1:

1) More detail needs to be given about acclimation times during the metabolic rate experiments. After shrimp were collected from the habitats and brought up, how long were they allowed to adjust to their new settings before metabolic rate measurements began at the different salinities? In most experiments of this kind, animals are allowed to settle in for a period of time before measurements begin (or before the data are recorded), because handling stress can cause MR to increase. In all the species, MRs are high initially at test salinities and then decrease over time. The authors interpret this as a physiological response to salinity, but it may be a result of handling stress during setting up the experiment and/or transfer of animals to the new salinity. If shrimp were brought up and then placed in the native salinity for control measurements over 150 minutes, the authors could address this by showing the same time series data (over 150 mins) for the control vs. test salinities to see if MRs were initially elevated there as well before lowering. Based on Fig 1, it seems this might not be the case, so the authors may not be able to disentangle time from salinity. In any case, stress due to handling and initial responses to the experimental apparatus (not salinity) should be discussed. The samples sizes for each measurement/experiment and species should also be referenced in the main text, not just Fig 1. 

Answer. We have expanded the methods section, describing the time elapsed from collection to experimentation, added the number of individuals used for each trial and a note stating that none were used more than once in any trial. We have also explained how we calculated the RMR in native conditions and added a phrase explaining why this was used as a reference for all other trials. Figure 2 has been modified and now reads: control (0-150 min).

The objective of these experiments was to characterize the physiological response of wild individuals in situ, without any acclimation to the confinement or more traditional experimental setups. Because we observed droppings and depletion of the digestive tract soon after Typhlatyas had been collected, we strived to begin the experiments as quickly as possible, to avoid experimenting with “hungry” individuals. 

Finally, handling stress was similar for individuals in all RMR trials, allowing for valid comparisons of the effect of the salinity on all indicators through time both in native (control) and challenging salinity scenarios. In particular, the RMR of control individuals was assumed to be constant throughout the 150 minutes, so any changes observed after being transferred to challenging scenarios could be attributed to salinity.

2) In interpreting the results, the authors should consider direction of salinity challenge as a possible explanation. T. mitchelli and T. pearsei were undergoing freshwater to seawater transfers and had increased stress/low survival. T. dzil. underwent an opposite transfer and had high survivorship. Given that the atyids are a freshwater family, increased salinity might be a natural stressor, while T. dzil. has switched to being a seawater specialist, but still maintains the ability to survive in freshwater. In truly euryhaline species, researchers often acclimate all species to the same salinity and then undergo a transfer in the same direction. While that isn’t possible in this system, it is worth noting that direction of transfer also covaried with tolerance here. It should also be noted that because of the nature of the system, an entirely controlled and comparable experiment is not possible.

Answer. We have added a paragraph to the discussion addressing the matter, emphasizing the freshwater nature of the atyids, the last common ancestor of the Yucatan-Cuba clade and how the difference of T. dzilamensis effectively colonizing the MG environment is not shared by its sister species. 

3) I found the presentation of the results somewhat confusing. This is especially true for figures 3 and 5. For fig 3 the main issue is that the PCA is difficult to interpret given the overlap of the points in many panels. Adding a mean and ellipse for each treatment would go a long way towards letting the reader see how the treatments are affecting the PCA values. I’d also suggest adding a set of supplementary figures similar to Fig2 and 4 for each of the enzymes examined, so readers interested in a particular enzyme can see how its activity changed with salinity in each species. Similarly, in Fig 5 the interesting comparison is how AS changes with salinity within each species, not differences between species at a particular salinity. I’d suggest reformatting this figure to match the layout in Figs. 2 and 4.

Answer. We insist on keeping Fig. 3 without adding ellipses. We believe this figure already displays two different responses: impacts regarding the magnitude of the salinity change (triangles and squares), and impacts derived by the time they are maintained at this new salinity (tones of blue). Furthermore, we show the enzymes profile in native conditions (green) without any salinity changes. We agree that this figure can be challenging because it condenses a lot of information. However, taking proper time to analyze this figure is important to understand how the changes in time and at different salinities have different impacts among these species and treatments. In hope of being clearer we have extended the description of these results and included a supplementary figure (SF1), where enzymes are shown for each species over time and salinity treatment, as suggested.

We had originally submitted fig 5 in this manner for consistency (matching the layout) with figs 2 and 4, and because the comparison within each species was simple. However, we agree with the suggestion of changing the order in fig 5.

4) Minor comments

1) Line 23 – “by the haloclines” should be “by haloclines”

Answer. Modified.

2) Line 32 – “novel physiological characteristics” is too vague. I suggest removing this and emphasizing that some species may be euryhaline while others are restricted to the freshwater portion of the habitat.

Answer. Modified.

3) Line 52 – remove “the” before “rapidly advancing climate change”

Answer. Removed.

4) Line 58 – change “is” to “are” and remove “the”

Answer. This has been modified and now reads “Acute changes in salinity are one of the defining characteristics of anchialine environments”

5) Line 113 – change “resource” to “resort”

Answer. Modified.

6) Line 136 – list collection parameters for T. pearsei here as well.

Answer. This has been modified and added to the Animal Collection subsection in Methods.

7) Line 190 – change “trough” to “through”

Answer. Modified

8) Line 225- change “essay” to “assay”

Answer. Modified

9) Line 309 – what statistics were used to analyze the aerobic scope data? I didn’t see this in the statistics section.

Answer. A new paragraph has been added explaining why only descriptive statistics were used to analyse the aerobic scope.

10) Line 319 – the laying out of all the different comparisons is quite confusing. I might suggest condensing this section and having a few lines for each species stating the general trend and then maybe the significant differences.

Answer. We have re-written the results section in order to articulate RMR values that were measured with the results of statistical procedures, and the observed behaviours of Typhlatya throughout trials. 

11) Prior to getting into the MR data, I’d suggest a few lines outlining the behavior and survival of each species instead of listing this later. This will give the reader an idea of general stress responses before getting into the MR data. In general, I think its ok to have a paragraph for each species where the behavior/survival observations are noted first, then the MR trends and significant differences. It is also difficult to keep track of the species if one isn’t familiar with them, so it is useful to reiterate for each species where they were collected (FW or SW) and which direction they were transferred in.

Answer. We have restructured the results section and hope these are more straightforward. 

12) Line 563 – this is a good point that I would like to see expanded upon – increased MR could be interpreted as an adaptive/beneficial response (they are able to mount a physiological response) or a maladaptive response (they are burning lots of energy). Because these interpretations are polar oppsites, its important to interpret the data here in both ways and discuss what either interpretation could mean for stress, etc.

Answer. We have changed the structure of the discussion and have included a paragraph in lines 643 – 676 which addresses this point.

13) Line 615: “which can”?

Answer. Deleted.

 

Reviewer #2:

Dear authors! I have reviewed your manuscript: “Are haloclines distributional barriers in anchialine ecosystems? Physiological response of cave shrimps to salinity.” It is a relevant and timely study bringing novel data that highlight potential issues certain species might experience in the rapidly changing world due to changes in salinity in a sensitive and rare environment. I did not find any major issues regarding the content but think that the paper is poorly organised; not clearly written and needs a major makeover of many parts. Please see my comments and suggestion below on how to improve this. 

INTRODUCTION

1) I think the introduction needs a thorough makeover. A substantial part of text (3rd, 4th, and 5th paragraph) is now very technical, devoted to introducing common, well-known physiological traits, their proxies and how to measure these. This has nothing to do with the story you are telling but is a much more related to the methods you used to assess sensitivity to salinity. I recommend you move most of this text to the appropriate parts of the methods section. In the intro, possibly just in the last paragraph, mention just the essentials. This will free-up space which I suggest you use to build a more solid story. I think you should make climate change (and anthropic) driven changes in environment and their effect on species living along haloclines and potential conservation issues as your main thread and strengthen it. 

Answer. We have restructured the introduction, moving most of the suggested paragraphs into a new the methods section (“Methodological background”), and we have modified our introduction to respond to environmental changes and their risk under current climate change projections.

2) Please include relevant recent review studies on threats to cave life and habitats like Mammola (many papers), Wynne et al. 2021, but also one of the landmark papers on anchialine cave ecology (Sket, 1996). Also, more specifically introduce what is known of salinity sensitivity/tolerance in cave animals in general. Due to increased threats of anthropic salination of freshwater, recently a few relevant studies were published (e.g., Castaño-Sánchez et al 2020, Jemec et al. 2022) and I suggest you refer to them either in introduction or devote a paragraph for this in discussion. Finally, please improve the presentation of your model system in the 6th paragraph. Say why are these shrimps a good model for your question, and let the reader know the essentials of the model system required to understand this. Sket, B 1996: https://doi.org/10.1016/0169-5347(96)20031-X; Mammola et al: https://doi.org/10.1111/brv.12642;
https://doi.org/10.1111/brv.12851;
https://doi.org/10.1093/biosci/biz064;
https://doi.org/10.1177/2053019619851594; Wynne et al. 2021: https://doi.org/10.1111/conl.12834; Castaño-Sánchez et al 2020: doi: 10.1038/s41598-020-69050-7; Jemec et al. 2022: https://doi.org/10.1016/j.ecoenv.2022.113456

Answer. We have included the suggested references in the introduction, and have devoted a paragraph in the discussion to salinity tolerance, as suggested. Additionally, we expanded the description of why we chose to work with Typhlatya species from Yucatan. 

3) First lines: support your claims with references.

Answer. Modified. 

4) Lines 54-57: first paragraph of the intro should introduce the broad topic; this interrupts the flow, please remove and reuse maybe in the last parts of the introduction; for example the 6th paragraph

Answer. We have re-structured the introduction as suggested and hope this has a better flow.

5) Lines 56-57: No need for abbreviations, in my opinion. Just use the full phrasing throughout the manuscript.

Answer. We insist on keeping this abbreviation in order to lighten the reading and avoid an excessively wordy manuscript. 

6) Lines 58-60: flip the order of the sentence

Answer. The phrase has been modified.

7) Line 63: FG and MG, usually your abbreviate as FGW and MGW.

Answer. This has been addressed throughout the manuscript.

8) Lines 68-69: impossible to understand without checking Box 1, but actually also not explained in the box, but in the 3rd paragraph of intro. Please rewrite so that the question is instantly understandable.

Answer. The glossary box now includes the proper terms and the phrase has been modified so that reader does not need to refer to the box.

9) Lines 71: no need for a capital letter in Oxygen

Answer. Modified.

10) Lines 78: a fullstop is missing after references

Answer. Modified.

11) Lines 85: start a new sentence after “decreases”

Answer. Modified.

12) Lines 86: rephrase “have exceeded”

Answer. This sentence has been modified and now reads “Negative AS values are only possible for short periods of time, as this implies that life maintenance costs exceed the ATP production capacity.”

13) Lines 90 & 95: lethal in italic

Answer. Modified.

14) Line 99: comma after demand

Answer. Modified.

15) Line 105: no need to use capital letters.

Answer. Modified.

16) Line 121: Please reconsider expressing your units of salinity (UPS) in the new standard, TEOS-10, so in promilles, g/kg, also dimensionless.

Answer. We have changed the PSU to the standard TEOS-10 for salinities from 2-35 and IES-80 for salinities under 2 PSU. We have stated the absolute salinity in the salinity scenarios and continued with the Ps (practical salinity) throughout the manuscript as a way to simplify the numbers for the readers. 

17) Lines 121-122: delete the sentence. It only adds confusion. Not clear what is this Yucatan-Cuba clade. Are Typhlatya part of it or not.

Answer. This has been modified and moved to the discussion section, as this information gives an evolutionary background to the Typhlatya species we use in this work. The sentence now reads: In particular, the Typhlatya pertaining to the Yucatan + Cuba clade were found to have a low salinity most recent common ancestor.

18) Lines 124: not clear what SGW is. Did you mean MGW?

Answer. Yes, thank you, this has been modified and now reads MG.

19) Lines 126-130: very long sentence and thus hard to comprehend. Consider breaking into two or three shorter sentences.

Answer. Sentence has been divided.

20) Lines 131: and should not be written in italic

Answer. Modified.

21) Lines132: not clear what you mean by “compared such consequences”. Rephrase.

Answer. We have modified to “compared such responses through…”

22) Line132: change metabolic rate to routine metabolic rate, otherwise the abbreviation does not match

Answer. Modified.

METHODS

Animal collection

1) It is not clear where T.pearsei was collected: in the solely freshwater cenotes? It is also not clear whether T. michelli was collected also from these two, or just from the Pandrosa system. I guess T.dzilamensis was collected only in the last system. From the next subchapter it seems you did not sample in the Tza Itza cenote at all. Why do you mention it here?

Answer. We have added the following sentence “. T. pearsei were collected from the cenote pool of the exclusively freshwater cenote Nohmozon at night.” And we have further explained that T mitchelli was collected from both Tza Itza and Ponderosa for the control individuals only.

2) Already in the introduction, you could say and explain more about the distribution of these shrimps and if and how often they co-occur. This relates to my previous comment in intro on improving model system descriptio

---

## [Editor Report · Decision Letter 1]

7 Jun 2024

Are haloclines distributional barriers in anchialine ecosystems? Physiological response of cave shrimps to salinity.

PONE-D-23-27494R1

Dear Dr. Mascaro,

We’re pleased to inform you that your manuscript has been judged scientifically suitable for publication and will be formally accepted for publication once it meets all outstanding technical requirements.

Kind regards,

Sanja Puljas

Academic Editor

PLOS ONE
---

## [Editor Report · Acceptance letter]

19 Jun 2024

PONE-D-23-27494R1 

PLOS ONE

Dear Dr. Mascaro, 

I'm pleased to inform you that your manuscript has been deemed suitable for publication in PLOS ONE. Congratulations! Your manuscript is now being handed over to our production team.

Kind regards, 

on behalf of

Dr. Sanja Puljas 

Academic Editor

PLOS ONE